

# Steroid hormones and chondrichthyan reproduction: physiological functions, scientific research, and implications for conservation

Edgar Eduardo Becerril-García[1], Marcial Arellano-Martínez[1],
Daniela Bernot-Simon[2], Edgar Mauricio Hoyos-Padilla[3,4],
Felipe Galván-Magaña[1] and Céline Godard-Codding[5]

[1] Instituto Politécnico Nacional, Centro Interdisciplinario de Ciencias Marinas, La Paz, Mexico
[2] Departamento Académico de Ciencias Marinas y Costeras, Universidad Autónoma de Baja California Sur, La Paz, Mexico
[3] Pelagios Kakunjá A.C., La Paz, Mexico
[4] Fins Attached, Colorado Springs, CO, USA
[5] The Institute of Environmental and Human Health, Texas Tech University, Lubbock, TX, USA

Corresponding author
Marcial Arellano-Martínez,
arellano.marcial@gmail.com

## ABSTRACT

The study of the reproductive aspects of chondrichthyans through the analysis of steroid hormones has been carried out for more than five decades in several species around the world. This scientific knowledge constitutes the basis of the reproductive endocrinology of chondrichthyans, which has provided information regarding their sexual maturation, gametogenesis, mating seasons, gestation periods, and parturition. The present review summarises the existing literature on steroid hormones in chondrichthyan reproduction and identifies future research directions addressing critical knowledge gaps in the reproductive physiology of this taxon. A total of 59 peer reviewed scientific papers from 1963 to 2020 were reviewed and the following parameters analysed: species, steroid hormones, biological matrix, field sampling (year, location), and methodology (assays, sample size, precision, and recoveries). We provided a summary of the methods, biological matrices, and the functions of up to 19 hormones on the biology of 34 species of chondrichthyans that have been analysed to date. The majority of the studies used radioimmunoassay as the main methodology (76.3%; $n = 45/49$); while the most frequent biological matrix used was plasma (69.5%; $n = 41/49$). A Kernel's heat map was generated to present the scientific effort according to geographic location and evidenced a lack of research in high biodiversity areas for chondrichthyans worldwide. The implications of the study of steroid hormones for the conservation of chondrichthyans are discussed, as only 2.9% of the species of this group have been analysed and most of the scientific effort (93.2%; $n = 55/59$ papers) has focused on the analysis of less than six hormones.

## INTRODUCTION

In recent decades, the study of hormones has greatly advanced our understanding of the reproductive biology of vertebrates. Most of the studies regarding steroid hormones in terrestrial and marine species have focused on three sex steroids: 17β-oestradiol (E2), testosterone (*T*) and progesterone (P4; *Idler, 2012*). These steroids are related to important reproductive processes such as spermatogenesis, oogenesis, the offset of sexual maturity and the maintenance of pregnancy in marine mammals, sea birds, and fishes (*Idler, 2012*; *Awruch, 2013*; *Hayden et al., 2017*). Analyses were performed primarily on captive or dead organisms, due to their invasive or lethal nature and the limitations of sampling free-ranging individuals. As a result, large knowledge gaps in reproductive physiology and endocrinology remain for several marine species, including vulnerable and ecologically important taxa such as sharks, rays, and chimaeras, collectively known as chondrichthyans (*Hammerschlag & Sulikowski, 2011*; *Maruska & Gelsleichter, 2011*; *Awruch, 2013*).

Chondrichthyans are characterised by the presence of a cartilaginous skeleton, the lack of a swim bladder and internal fertilisation (*Compagno, Dando & Fowler, 2005*). The class Chondrichthyes includes more than 509 species of sharks, 630 rays and 49 chimaeras and comprises oviparous, as well as placental or aplacental viviparous species (*Hamlett, 2011*; *Castro, Sato & Bodine, 2016*; *Weigmann, 2016*). The range of reproductive biology strategies observed in this high diversity of species and associated habitats reflects different metabolic processes that have evolved for more than 400 million years (*Compagno, Dando & Fowler, 2005*; *Maruska & Gelsleichter, 2011*; *Gelsleichter & Evans, 2012*; *Castro, Sato & Bodine, 2016*).

Information provided by endocrinology studies regarding the mechanisms of sexual maturity, mating, gestation and parturition seasons of chondrichthyan populations, particularly for species relevant to fisheries and ecotourism, is essential for the economy of several countries around the world (*Awruch, 2013*; *Cisneros-Montemayor et al., 2020*). The present review provides practical information regarding the steroid hormones studied in chondrichthyans, with a focus on the number of species and the type of reproduction, methodologies, biological matrices, and steroid panels analysed to date. Additionally, the scientific effort and the implications of steroids for the conservation of chondrichthyans are discussed to provide insights about new alternatives for the study of hormones in protected and non-protected species.

## SURVEY METHODOLOGY

Literature related to steroid hormones in the reproduction of chondrichthyans was obtained from several sources including Google Scholar, Web of Science, university and research centre websites, as well as scientific journal libraries using key words such as steroid, hormone, reproduction, and shark, ray or chimaera. From this search, a total of 1,630 results were obtained and checked for suitability according to the aims of the present review. Only peer reviewed papers were considered, resulting in 59 scientific articles and short communications that were published between 1963 and 2020. Data from each paper was extracted in order to classify the species involved, sex, analysed hormones, physiological function, year of study, location, tissue or biological matrix, used method,

original volume per sample, body area for sample collection, sample size and reported recoveries. These data were presented in histograms while the location and scientific effort were analysed using a Kernel distribution heat map with the software QGIS v. 3.10.2. Additionally, the species reproductive mode, taxonomy and conservation status were determined according to *Compagno, Dando & Fowler (2005)*, *Castro, Sato & Bodine (2016)*, *Froese & Pauly (2019)* and the IUCN Red List of Threatened Species (2020).

## Analysed species

Reproductive steroid hormones analyses were reported in a total of 34 chondrichthyans, including 21 sharks, 12 batoids and one chimaeriform (Table 1). Half of the reviewed studies included both males and females in their analyses (45.8%; $n = 27$), while 37.3% ($n = 22$) described only females and 16.9% ($n = 10$) solely males. This scientific effort spanned the description of diverse reproductive strategies observed in chondrichthyans, including 22 viviparous (64.7%) and 12 oviparous species (35.3%; Fig. 1A).

Embryonic nutrition is related to the reproductive cycle of viviparous elasmobranchs and explains the varied endocrinology observed within this taxon (*Maruska & Gelsleichter, 2011*). Embryos of viviparous species could complete their development through a placental connection, ingestion of unfertilised oocytes or other embryos, yolk sac, or nutritional substances provided in histotrophic and embryotrophic species (*Compagno, Dando & Fowler, 2005*; *Hamlett, 2011*; *Castro, Sato & Bodine, 2016*; *Nelson, Grande & Wilson, 2016*). However, the number of studies is highly limited in certain groups, with few papers published for some reproductive modes such as oophagy (*Carcharodon carcharias*; *Sulikowski, Williams & Domeier, 2012*), embryotrophy (*Galeocerdo cuvier*; *Sulikowski et al., 2016*) and adelphophagy (*Carcharias taurus*; *Henningsen et al., 2008*; *Wyffels et al., 2019*).

Sex steroids have been studied in oviparous chondrichthyans since 1979, including in six species of sharks, five species of rays and the spotted ratfish *Hydrolagus colliei* as the only chimaera (Table 1). The 18 papers published to date constitute the basis of the scientific knowledge of the reproductive endocrinology of oviparous chondrichthyans, providing some insights regarding capsule protein synthesis, sperm storage and oviposition (*Sumpter & Dodd, 1979*; *Awruch et al., 2009*; *Barnett et al., 2009*; *Nau et al., 2018*).

To date, there is no general pattern that could describe the specific actions of sex steroids in chondrichthyans, mainly due to the high diversity of species in this group and the few number of comprehensive and detailed research involving specimens in captivity (*Anderson et al., 2018*). The lack of a general pattern is probably due to the variety of embryonic nutrition strategies presented in the group, the overall paucity of species investigated, as well as inherent differences between species, populations, and individuals (*Compagno, Dando & Fowler, 2005*; *Awruch, 2013*; *Nelson, Grande & Wilson, 2016*). In a general perspective, sex steroids have been described in only 2.9% of the 1,188 living species of chondrichthyans (*Weigmann, 2016*). Although previous studies have allowed the partial description of the endocrinology in multiple reproductive strategies (Table 1), the proportion of species analysed to date for their steroid hormones is scarce compared

**Table 1 Chondrichthyan species subject to steroid hormones and reproduction analyses during 1963–2020.** Data include reproductive strategy and the conservation status according to *IUCN (2020)*.

| Species | Common name | Order | Family | Reproductive strategy | IUCN status |
|---|---|---|---|---|---|
| *Hydrolagus colliei* | Spotted ratfish | Chimaeriformes | Chimaeridae | Oviparous | LC |
| *Notorynchus cepedianus* | Broadnose sevengill | Hexanchiformes | Hexanchidae | Yolk-sac viviparous | DD |
| *Centroscymnus coelolepis* | Portuguese dogfish | Squaliformes | Somniosidae | Yolk-sac viviparous | NT |
| *Squalus acanthias* | Spiny dogfish shark | Squaliformes | Squalidae | Yolk-sac viviparous | VU |
| *Chiloscyllium plagiosum* | White-spotted bamboo shark | Orectolobiformes | Hemiscylliidae | Oviparous | NT |
| *Hemiscyllium ocellatum* | Eupaulette shark | Orectolobiformes | Hemiscylliidae | Oviparous | LC |
| *Rhincodon typus* | Whale shark | Orectolobiformes | Rhincodontidae | Yolk-sac viviparous | EN |
| *Stegostoma fasciatum* | Zebra shark | Orectolobiformes | Stegostomatidae | Oviparous | EN |
| *Carcharodon carcharias* | White shark | Lamniformes | Lamnidae | Oophagy | VU |
| *Carcharias taurus* | Sand tiger shark | Lamniformes | Lamnidae | Adelphophagy | VU |
| *Carcharhinus leucas* | Bull shark | Carcharhiniformes | Carcharhinidae | Placental viviparous | NT |
| *Carcharhinus plumbeus* | Sandbar shark | Carcharhiniformes | Carcharhinidae | Placental viviparous | VU |
| *Galeocerdo cuvier* | Tiger shark | Carcharhiniformes | Carcharhinidae | Embryotrophy | NT |
| *Negaprion brevirostris* | Lemon shark | Carcharhiniformes | Carcharhinidae | Placental viviparous | NT |
| *Prionace glauca* | Blue shark | Carcharhiniformes | Carcharhinidae | Placental viviparous | NT |
| *Rhizoprionodon taylori* | Australian sharpnose shark | Carcharhiniformes | Carcharhinidae | Placental viviparous | LC |
| *Rhizoprionodon terranovae* | Atlantic Sharpnose shark | Carcharhiniformes | Carcharhinidae | Placental viviparous | LC |
| *Cephalloscyllium laticeps* | Draughtboard shark | Carcharhiniformes | Scyliorhinidae | Oviparous | LC |
| *Scyliorhinus canicula* | Small spotted catshark | Carcharhiniformes | Scyliorhinidae | Oviparous | LC |
| *Scyliorhinus stellaris* | Nursehound | Carcharhiniformes | Scyliorhinidae | Oviparous | NT |
| *Sphyrna tiburo* | Bonnethead shark | Carcharhiniformes | Sphyrnidae | Placental viviparous | LC |
| *Mustelus schmitti* | Narrownose smooth hound | Carcharhiniformes | Triakidae | Placental viviparous | EN |
| *Trygonorrhina dumerilii* | Southern fiddler ray | Rhinopristiformes | Trygonorrhinidae | Yolk-sac viviparous | LC |
| *Amblyraja radiata* | Thorny skate | Rajiformes | Rajidae | Oviparous | VU |
| *Leucoraja erinacea* | Little skate | Rajiformes | Rajidae | Oviparous | NT |
| *Leucoraja ocellata* | Winter skate | Rajiformes | Rajidae | Oviparous | EN |
| *Malacoraja senta* | Smooth skate | Rajiformes | Rajidae | Oviparous | EN |
| *Raja eglanteria* | Clearnose skate | Rajiformes | Rajidae | Oviparous | LC |
| *Torpedo marmorata* | Spotted ray | Torpediniformes | Torpedinidae | Histotrophic viviparous | DD |
| *Hypanus sabinus* | Freshwater Atlantic stingray | Myliobatiformes | Dasyatidae | Histotrophic viviparous | LC |
| *Hypanus americanus* | Southern stingray | Myliobatiformes | Dasyatidae | Histotrophic viviparous | DD |
| *Mobula alfredi* | Reef manta ray | Myliobatiformes | Myliobatidae | Histotrophic viviparous | VU |
| *Rhinoptera bonasus* | Cownose rays | Myliobatiformes | Myliobatidae | Histotrophic viviparous | NT |
| *Urobatis halleri* | Round stingray | Myliobatiformes | Urotrygonidae | Histotrophic viviparous | LC |

**Note:**
DD, Data Deficient; LC, Least Concern; NT, Near Threatened; VU, Vulnerable; EN, Endangered.

to the ecological richness of this taxonomic class. For instance, only 4.1% of the total of sharks, 1.9% of batoids, and 2.0% of chimaeras are represented (*Weigmann, 2016*).

The majority of the analysed taxa constitute marine resources for artisanal and industrial fisheries in several localities around the globe (*Compagno, Dando & Fowler, 2005*; *Nelson, Grande & Wilson, 2016*). The use of commercial species as biological models

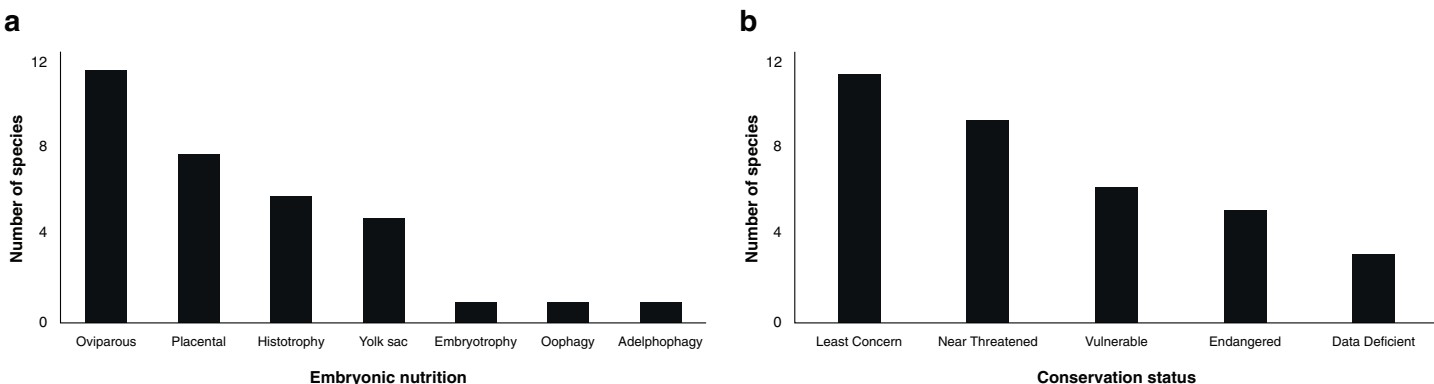

**Figure 1** Number of chondrichthyans according to embryonic nutrition (A) and conservation status (B; *IUCN, 2020*) that have been studied in terms of steroid hormones during 1963–2020 (*n* = 34 species).

has allowed the analysis of both gonads and blood, as well as the improvement of methods for the detection of steroid hormones. Elasmobranch species listed as 'Least Concern', such as the small-spotted catshark *Scyliorhinus canicula*, the bonnethead shark *Sphyrna tiburo* and the round stingray *Urobatis halleri* are examples of such biological models (Table 2). The large number of steroid hormones analysed in these chondrichthyan species improved our understanding of the reproductive endocrinology of this group, particularly in terms of sexual maturation, gametogenesis and gestation (*Sumpter & Dodd, 1979*; *Manire, Rasmussen & Gross, 1999*; *Mull, Lowe & Young, 2008*; *Lyons & Wynne-Edwards, 2019*).

Analyses involving lethal methods on commercial species has partially been used as a comparison or baseline when studying threatened species. In this regard, a third of the analysed chondrichthyans are considered vulnerable or endangered according to the *IUCN (2020)*, although this status is not always associated with their protection (Fig. 1B). Some of the analysed species pertaining to economic activities other than fisheries such as ecotourism are the white shark *C. carcharias*, the whale shark *Rhincodon typus* and the reef manta ray *Mobula alfredi*. However, only the study of *C. carcharias* carried out by *Sulikowski, Williams & Domeier (2012)* has involved the description of hormonal levels in a wild population through invasive practices, while the studies of whale sharks and manta rays have been made in captivity (*Nozu et al., 2015*, *2017*; *Matsumoto et al., 2019*). Even though these and other species are relevant for ecotourism worldwide, available information about their reproduction remains scarce for numerous populations (*Huveneers et al., 2018*; *Cisneros-Montemayor et al., 2020*).

## Methods and biological matrices

The study of steroid hormones in the reproduction of chondrichthyans has been carried out using five analytical methods (Fig. 2A) that can be grouped in two main categories: (1) immunoassay-based including radioimmunoassay (RIA), enzyme immunoassay or enzyme-linked immunosorbent assay (EIA/ELISA), and time-resolved fluorescent immunoassay (TRFIA); and (2) physical separation-based including paper chromatography or thin layer chromatography followed by gas chromatography

**PeerJ** ____________________________________________________________________

**Table 2 Scientific research regarding steroid hormones in chondrichthyan reproduction during 1963–2020.** Data includes methods (RIA: radioimmunoassay; EIA/ELISA: enzyme immunoassay or enzyme-linked immunosorbent assay; TRFIA: time resolved fluorescent assay; PC: paper chromatography; TLC-GC: thin-layer chromatography followed by gas chromatography; LC-MS/MS: Liquid chromatography tandem mass spectrometry); tissues (S: semen; P: plasma; Se: serum; G: gonad; Y: yolk; M: muscle; H: histotroph); analysed hormones (E2, 17β-oestradiol; T, testosterone; P4, progesterone; DHT, 5α-dihydrotestosterone; 11-KT, 11-ketotestosterone; CORT, corticosterone; A4, androstenedione; 17-OHP, 17-hydroxyprogesterone; E1, oestrone; E3, oestriol; E, cortisone; F, cortisol; S, 11-deoxycortisol; DHP, dihydroprogesterone; DOC, 11 deoxycorticosterone; 11KA4, 11-ketoandrostenedione; 3α-diol, 5α-androstane-3α,17β-diol; 17P5, 17 hydroxypregnenolone; 11-DHC, 11-dehydrocorticosterone); original volume per sample (g, mL); volume for analysis (µL); recovery or extraction efficiency (%); and lethality of the method. Dash (–) used when the information was not specified.

| Species | Method | Tissue | Analysed hormones | Volume per sample | Volume for analysis | Extraction efficiency | Lethality | References |
|---|---|---|---|---|---|---|---|---|
| *Hydrolagus colliei* | RIA | M, P | E2, T, 11-KT | – | 0.5 g, 500 µL | 69–111 | Lethal | *Barnett et al. (2009)* |
| *Notorhynchus cepedianus* | RIA | P | E2, T, P4 | 3 mL | 200 µL | 83–92 | Lethal | *Awruch et al. (2014)* |
| *N. cepedianus* | RIA | P | E2, T, P4 | 5 mL | 200 µL | – | Lethal | *Sueiro et al. (2019)* |
| *Centroscymnus coelolepis* | RIA | P | E2, T, P4 | 1–1.5 mL | 25 µL | – | Lethal | *Tosti et al. (2006)* |
| *Squalus acanthias* | RIA | P | E2, T, P4 | 0.5–1 mL | 50 µL | 80–92 | Lethal | *Tsang & Callard (1987)* |
| *S. acanthias* | RIA | P | E2, T, P4 | 5 mL | – | 67–82 | Lethal | *Bubley et al. (2013)* |
| *S acanthias* | RIA | M, G, P | E2, T, P4 | 5 g, 8 mL | 2g, 100 µL | 20–85 | Lethal | *Prohaska et al. (2013a)* |
| *S. acanthias* | RIA | M, P | E2, T, P4 | 5 g, 8 mL | 2g, 100 µL | 74–91 | Lethal | *Prohaska et al. (2013b)* |
| *S. acanthias* | RIA | P | E2, T, P4 | 5 mL | 400 µL | 71–90 | Lethal | *Prohaska et al. (2018)* |
| *Chiloscyllium plagiosum* | RIA | P | E2, T, P4 | 2.5 mL | 200 µL | 17–93 | Non-Lethal | *Nau et al. (2018)* |
| *Hemiscyllium ocellatum* | RIA | P | E2, P4, T | 1 mL | – | – | Lethal | *Heupel, Whittier & Bennett (1999)* |
| *Carcharodon carcharias* | RIA | P | E2, T, P4 | 10 mL | 400 µL | – | Non-Lethal | *Sulikowski, Williams & Domeier (2012)* |
| *Carcharias taurus* | RIA | Se | E2, T, P4, DHT | 10 mL | – | 66–81 | Non-Lethal | *Henningsen et al. (2008)* |
| *Carcharhinus leucas* | RIA | Se | E2, T, P4, DHT | 5 mL | – | – | Non-Lethal | *Rasmussen & Murru (1992)* |
| *Carcharhinus plumbeus* | RIA | Se | E2, T, P4, DHT | 5 mL | – | – | Non-Lethal | *Rasmussen & Murru (1992)* |
| *Galeocerdo cuvier* | RIA | P | E2, T, P4 | 20 mL | 500 µL | 59–84 | Non-Lethal | *Sulikowski et al. (2016)* |
| *Negaprion brevirostris* | RIA | Se | E2, T, P4, DHT | 5 mL | – | – | Non-Lethal | *Rasmussen & Murru (1992)* |
| *N. brevirostris* | RIA | Se | E2, T, P4, DHT, CORT | 5–20 mL | – | – | Non-Lethal | *Rasmussen & Gruber (1993)* |
| *Rhizoprionodon taylori* | RIA | P | E2, T, P4 | 3 mL | 100 µL | 86–90 | Lethal | *Waltrick et al. (2014)* |
| *Rhizoprionodon terranovae* | RIA | M, G, P | E2, T, P4 | 5 g, 8 mL | 2g, 100 µL | 39–87 | Lethal | *Prohaska et al. (2013a)* |
| *R. terranovae* | RIA | P, M | E2, T, P4 | 5 g, 8 mL | 2g, 100 µL | 74–91 | Lethal | *Prohaska et al. (2013b)* |
| *R. terranovae* | RIA | E2, T, P4P | | 5 mL | 400 µL | 60–91 | Lethal | *Prohaska et al. (2018)* |
| *Cephalloscyllium laticeps* | RIA | P | E2, T, P4, 11-KT | 3 mL | 200 µL | 74–88 | Lethal | *Awruch et al. (2008b)* |
| *C. laticeps* | RIA | P | E2, T, P4 | 3 mL | 200 µL | 74–86 | Lethal | *Awruch et al. (2008a)* |

| Species | Method | Tissue | Analysed hormones | Volume per sample | Volume for analysis | Extraction efficiency | Lethality | References |
|---|---|---|---|---|---|---|---|---|
| *C. laticeps* | RIA | *P* | E2, *T*, P4 | 3 mL | 200 µL | 74–86 | Lethal | *Awruch et al. (2009)* |
| *Scyliorhinus canicula* | RIA | *P* | E2, *T* | 2 mL | 50 µL | – | Lethal | *Sumpter & Dodd (1979)* |
| *S. canicula* | RIA | *P, G* | E2, *T*, P4, DHT, 11-KT, E1, A4, 17-OHP, 3α-diol | – | – | 35–85 | Lethal | *Garnier, Sourdaine & Jégou (1999)* |
| *Sphyrna tiburo* | RIA | Se | E2, *T*, P4, DHT | – | 500 µL | 70–88 | Lethal | *Manire et al. (1995)* |
| *S. tiburo* | RIA | Se | E2, *T*, P4, DHT | – | 500 µL | – | Lethal | *Manire & Rasmussen (1997)* |
| *S. tiburo* | RIA | Se | E2, *T*, P4, DHT, 11-KT, DHP, 11KA4 | – | 50 µL | 84–87 | Lethal | *Manire, Rasmussen & Gross (1999)* |
| *S. tiburo* | RIA | Se | E2, *T*, P4, DHT | – | 500 µL | – | Lethal | *Gelsleichter et al. (2002)* |
| *S. tiburo* | RIA | Se, Y | E2, *T*, P4 | – | 500 µL | 62–68 | Lethal | *Manire et al. (2004)* |
| *S. tiburo* | RIA | *P* | CORT | – | 25–75 µL | 89 | Lethal | *Manire et al. (2007)* |
| *Mustelus schmitti* | RIA | *P* | E2, *T*, P4 | – | 500 µL | 95 | Lethal | *Elisio et al. (2019)* |
| *Trygonorrhina dumerilii* | RIA | *P* | E2, *T*, P4 | 2 ml | 200 µL | 82–93 | Non-Lethal | *Guida et al. (2017)* |
| *Amblyraja radiata* | RIA | *P* | E2, *T* | 5–10 mL | 400 µL | 68–77 | Lethal | *Sulikowski et al. (2006)* |
| *A. radiata* | RIA | *P* | E2, *T*, P4 | 5–10 mL | – | 68–76 | Lethal | *Kneebone et al. (2007)* |
| *A. radiata* | RIA | *P* | E2, *T* | 5–10 mL | – | – | Lethal | *Sulikowski et al. (2007)* |
| *Leucoraja erinacea* | RIA | *P* | E2, *T*, P4 | – | 500 µL | 81–94 | Non-Lethal | *Koob, Tsang & Callard (1986)* |
| *L. erinacea* | RIA | *M, P* | E2, *T*, P4 | 5 g, 8 mL | 2g, 100 µL | 74–91 | Lethal | *Prohaska et al. (2013b)* |
| *L. erinacea* | RIA | *P* | E2, P4 | 5 mL | – | 75–76 | Non-Lethal | *Williams et al. (2013)* |
| *Leucoraja ocellata* | RIA | *P* | E2, *T*, P4 | 5–10 mL | 400 µL | 68–76 | Lethal | *Sulikowski, Tsang & Howell (2004)* |
| *L. ocellata* | RIA | *P* | E2, *T* | 5–10 mL | 400 µL | 74–76 | Lethal | *Sulikowski, Tsang & Howell (2005)* |
| *L. ocellata* | RIA | *P* | E2, *T* | 5–10 mL | 400 µL | – | Lethal | *Sulikowski et al. (2007)* |
| *Malacoraja senta* | RIA | *P* | E2, *T*, P4 | 5–10 mL | – | 77–88 | Lethal | *Kneebone et al. (2007)* |
| *Raja eglanteria* | RIA | *P* | E2, *T*, P4, DHT | 1–2 ml | 500 µL | 70–88 | Non-Lethal | *Rasmussen, Hess & Luer (1999)* |
| *T. marmorata* | RIA | *P, G, H* | E2, *T*, P4, DHT | – | – | 85–90 | Lethal | *Fasano et al. (1992)* |
| *Hypanus sabinus* | RIA | *P* | E2, *T*, P4, DHT, CORT | 5-10 mL | 500 µL | 70–78 | Lethal | *Snelson et al. (1997)* |
| *H. sabinus* | RIA | *P* | E2, *T*, P4, DHT | – | 500 µL | 70–88 | Lethal | *Tricas, Maruska & Rasmussen (2000)* |
| *H. sabinus* | RIA | *P* | E2, *T* | 5 mL | 500 µL | >78 | Lethal | *Gelsleichter et al. (2006)* |
| *H. sabinus* | RIA | *P* | CORT | – | 25–75 µL | 89 | Lethal | *Manire et al. (2007)* |
| *Rhinoptera bonasus* | RIA | *P* | E2, *T*, P4, A4 | 1–3 ml | – | 95–110 | Non-Lethal | *Sheldon et al. (2018)* |
| *Urobatis halleri* | RIA | *P* | *T*, 11-KT | – | – | – | Lethal | *Mull, Lowe & Young (2008)* |
| *U. halleri* | RIA | *P* | E2, *T*, P4 | – | – | – | Lethal | *Mull, Lowe & Young (2010)* |

(Continued)

| Species | Method | Tissue | Analysed hormones | Volume per sample | Volume for analysis | Extraction efficiency | Lethality | References |
|---|---|---|---|---|---|---|---|---|
| *Stegostoma fasciatum* | EIA/ELISA | P | E2, *T*, P4 | – | – | – | Non-Lethal | *Nozu et al. (2018)* |
| *Rhincodon typus* | EIA/ELISA | Se | E2, *T*, DHT | – | – | – | Lethal | *Nozu et al. (2015)* |
| *C. taurus* | EIA/ELISA | P | *T* | 15 mL | 50 µL | – | Non-Lethal | *Wyffels et al. (2019)* |
| *Prionace glauca* | EIA/ELISA | P | P4, E2 | 5 mL | – | – | Lethal | *Fujinami & Semba (2020)* |
| *R. terranovae* | EIA/ELISA | P | E2, *T* | 2 mL | – | – | Non-Lethal | *Hoffmayer et al. (2010)* |
| *T. marmorata* | EIA/ELISA | P | E2, P4 | – | 100 µL | 85 | Lethal | *Prisco et al. (2008)* |
| *Hypanus americanus* | EIA/ELISA | P | E2, *T*, P4, E1 | – | 50 µL | 94–108 | Non-Lethal | *Mylniczenko et al. (2019)* |
| *Mobula alfredi* | EIA/ELISA | P | E2, *T*, P4, DHT | – | – | – | Non-Lethal | *Nozu et al. (2017)* |
| *R. typus* | TRFIA | P | P4, E2 | – | – | – | Non-Lethal | *Matsumoto et al. (2019)* |
| *S. acanthias* | PC | S | DOC | – | – | – | Lethal | *Simpson, Wright & Gottfried (1963)* |
| *S. acanthias* | PC | S | DOC | 35 g | – | 65 | Lethal | *Simpson, Wright & Hunt (1964)* |
| *Scyliorhinus stellaris* | TLC-GC | S | *T*, 17P5 | 19 g | – | 34 | – | *Gottfried & Chieffi (1967)* |
| *Torpedo marmorata* | TLC-GC | P | *T*, P4, CORT, *E*, *F*, DOC, E1, E3 | 5–12 mL | – | – | Lethal | *Di Prisco, Vellano & Chieffi (1967)* |
| *U. halleri* | LC-MS/MS | P, H | E2, *T*, P4, 11-KT, *E*, CORT, *F*, 17-OHP, S, E1, E3, A4, 11-DHC | 10 mL | 50 µL | – | Lethal | *Lyons & Wynne-Edwards (2019)* |

(PC/TLC-GC), and liquid chromatography coupled with triple quadrupole tandem mass spectrometry and electrospray ionisation with multiple reaction monitoring (LC-MS/MS). In the case of immunoassays, a diverse number of antibodies have been used, particularly polyclonal full serum antibodies raised in mammals or amphibians for E2, *T*, and P4 (*Tosti et al., 2006*; *Awruch et al., 2009*; *Hoffmayer et al., 2010*; *Nau et al., 2018*). However, antibodies for relevant steroids such as 1α-hydroxycorticosterone were not available at the time of study (*Rasmussen & Gruber, 1993*). A brief description of the methods used and their applications in chondrichthyans is presented below:

### RIA

Assay based on antigen-antibody complex and radioactively labelled antigens (*Goldsmith, 1975*). This immunoassay methodology was first carried out in chondrichthyans by *Sumpter & Dodd (1979)* using the plasma of *S. canicula*. Since then, RIA have analysed an average of 3.36 (S.D. ± 1.31) steroid hormones per paper (Fig. 3) with hormone recoveries ranging from 17% to 111% (Table 2). Minimum and maximum number of specimens were 2–248 for males (*Tricas, Maruska & Rasmussen, 2000*; *Sulikowski et al., 2016*) and 1–248 for females (*Rasmussen & Murru, 1992*; *Tricas, Maruska & Rasmussen, 2000*). From the reviewed studies, only 61.7% specified the polyclonal antibodies used.

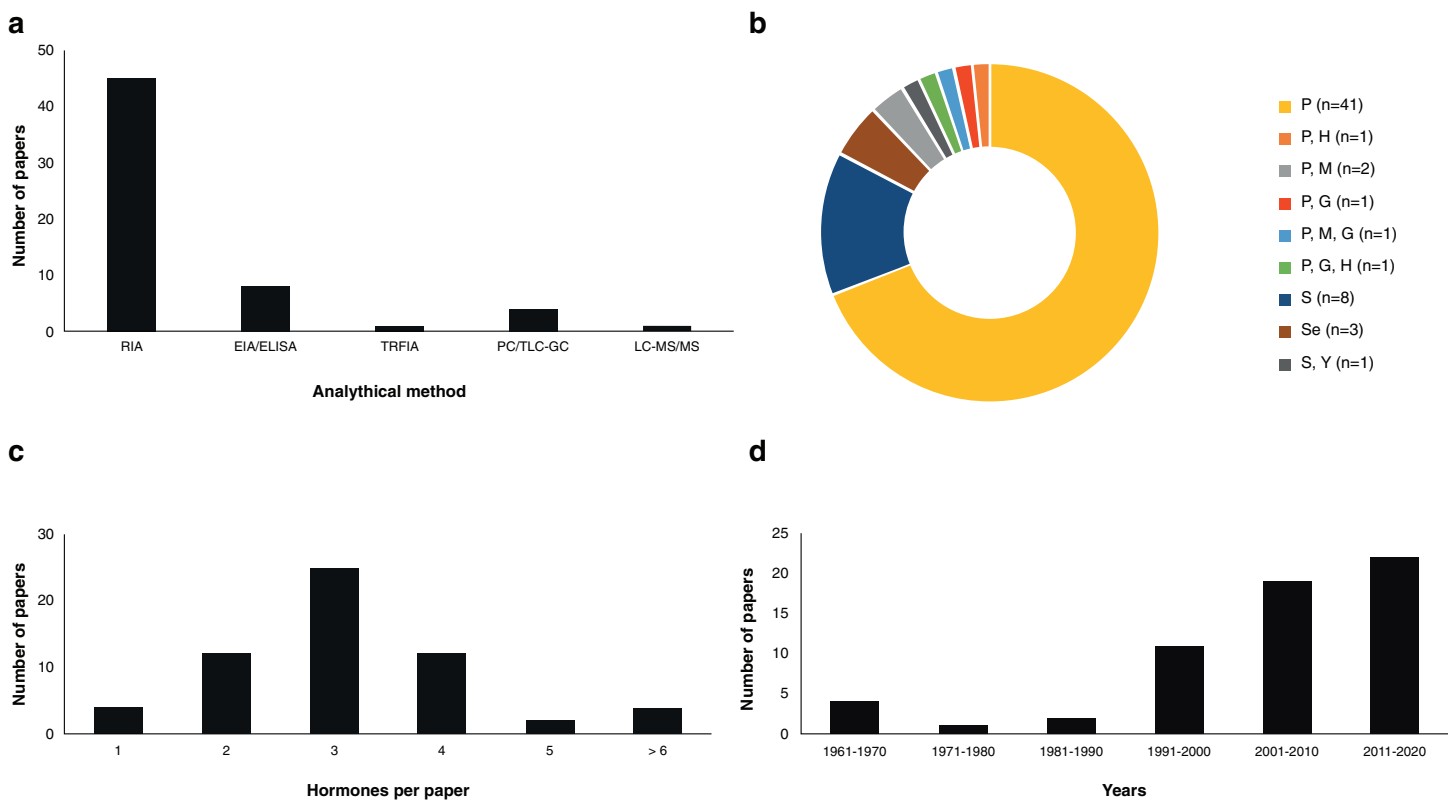

**Figure 2 Scientific literature related to the analysis of steroid hormones in chondrichthyans during 1963–2020 ($n$ = 59 papers).** (A) Methodology used, (B) biological matrix, (C) hormones per paper, and (D) number of papers according to decade. Abbreviations: RIA, Radio-immunoassay; EIA/ELISA, Enzyme immunoassay or enzyme-linked immunosorbent assay; TRFIA, Time-resolved fluorescent immunoassay; PC/TLC-GC, Paper chromatography or thin layer chromatography followed by gas chromatography; LC-MS/MS, Liquid chromatography coupled with triple quadrupole tandem mass spectrometry and electrospray ionisation with multiple reaction monitoring; P, Plasma; H, Histotroph; M, Muscle; G, Gonad; S, Serum; Se, Semen; Y, Yolk.

A total of 28 species of chondrichthyans have been analysed through RIA, including 17 sharks, 10 rays and one chimaera (Table 2). This biochemical assay was used in 76.3% ($n$ = 45) of the reviewed studies, making RIA the most widely used technique for the study of steroid hormones in these group of cartilaginous fishes (Fig. 2A).

### *EIA/ELISA*

Biochemical assay techniques focused on the detection and quantification of a wide range of substances including antibodies, allergens, proteins, peptides, hormones, etc. (*Vashist & Luong, 2018*). These biochemistry assays were first used to describe steroid hormones in the plasma of the spotted ray *Torpedo marmorata* (*Prisco et al., 2008*), and subsequently in the Atlantic sharpnose shark *Rhizoprionodon terranovae* (*Hoffmayer et al., 2010*). Since then, it has been used for the study of captive endangered species by analysing semen of the whale shark *R. typus* (*Nozu et al., 2015*) and plasma of the reef manta ray *M. alfredi* (*Nozu et al., 2017*) and zebra shark *Stegostoma fasciatum* (*Nozu et al., 2018*). The studies that have applied EIA/ELISA detected an average of 2.62 (S.D. ± 1.06) steroid hormones per paper with recoveries of 85–108% (Fig. 3). Minimum and maximum number of specimens were 2–81 for males (*Nozu et al., 2017*; *Wyffels et al., 2019*) and

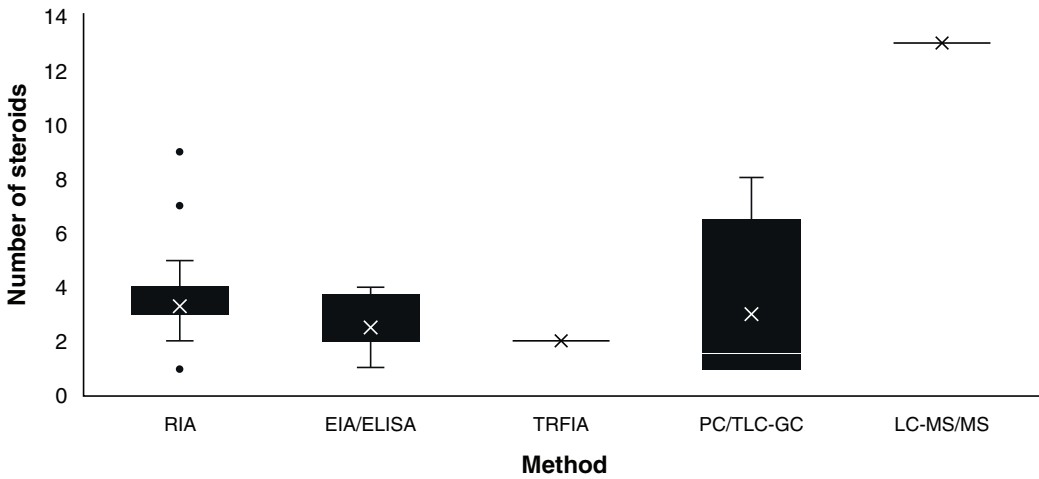

**Figure 3 Number of steroids per paper according to the method used for the study of chondrichthyan reproduction during 1963–2020.** Mean, 25% and 75% percentiles, outliers (●), minimum and maximum (*n* = 59 papers). Abbreviations: RIA, Radioimmunoassay; EIA/ELISA, Enzyme immunoassay or enzyme-linked immunosorbent assay; TRFIA, Time-resolved fluorescent immunoassay; PC/TLC-GC, Paper chromatography or thin layer chromatography followed by gas chromatography; LC-MS/MS, Liquid chromatography coupled with triple quadrupole tandem mass spectrometry and electrospray ionisation with multiple reaction monitoring.

3–94 for females (*Nozu et al., 2015*; *Mylniczenko et al., 2019*). All papers with EIA/ELISA as the main analytical method mentioned the antibodies used and constitute 13.6% (*n* = 8) of the reviewed literature.

### TRFIA

This non-radioactive immunoassay involves the use of Europium labelled antibodies and antigens for the detection of substances along with a fluorometer for the quantification through means of lanthanide fluorescence (*Diamandis, 1988*). In chondrichthyans it was used by *Matsumoto et al. (2019)* for the analysis of T and P4 in plasma of a captive whale shark. The monitoring for 20 years of this young male allowed to describe the effect of T and P4 on the sexual maturation of this endangered species, since an increase of T and P4 was related to the morphogenesis and functioning of the claspers. The antibodies used in this article were not specified. This study constitutes 1.7% of the reviewed papers.

### PC/TLC-GC

Physical techniques for the separation of components from a specific solution trough a stationary and a moving phase, which is paper and a liquid solvent in PC and silica plates interacting with liquids or gas in TLC and GC, respectively (*Bhawani et al., 2010*). PC was first reported for steroid hormones in chondrichthyans by *Simpson, Wright & Gottfried (1963)* and *Simpson, Wright & Hunt (1964)* in semen of *S. canicula*. In the study of *Di Prisco, Vellano & Chieffi (1967)*, the analysis of plasma of *T. marmorata* allowed the detection of eight sex steroid by using TLC-GC (Table 2). The studies that have required this technique have analysed an average of 3.36 (S.D. ± 1.68) steroids per paper (Fig. 3). Recoveries of steroid hormones ranged from 34% to 65% depending on the author (Table 2). Minimum and maximum number of specimens were 5–40 for males

(*Simpson, Wright & Hunt, 1964*; *Gottfried & Chieffi, 1967*) and 38 for females (*Di Prisco, Vellano & Chieffi, 1967*). These methodologies were mostly used during the sixties and constitute 6.8% (*n* = 4) of the reviewed studies.

### LC-MS/MS

This method combines chromatography and mass spectrometry for the separation and quantification of a wide range of substances including most organic compounds (large proteins, pollutants, drugs, and hormones; *Wudy et al., 2018*). It was recently used by *Lyons & Wynne-Edwards (2019)* for the simultaneous analysis of 13 steroid hormones in plasma and histotroph samples of the round stingray *U. halleri*. While some of these steroids were below the limit of quantification for selected samples, this was the study with the highest number of detected hormones overall. However, this paper did not report recoveries of the extracted steroids from the tissues of the 55 female specimens analysed. This technique was used only in this study and constitutes 1.7% of the reviewed papers.

The above methodologies used for the detection and quantification of steroid hormones were applied to six biological matrices: blood (plasma or serum), semen, yolk, histotroph, gonads and muscle. Most studies were performed using plasma (69.5%; *n* = 41 papers) or serum (13.6%; *n* = 8 papers), with three studies including semen (5.1%; *n* = 3 papers) and seven papers involving the analysis of other matrices (11.9%; Fig. 2B). In the case of blood, the original volume per sample ranged between 0.5 and 20 mL, while the volume for analysis was 25–1,000 μL (Table 2). Other tissues such as muscle or gonad required approximately 3–5 g of the original sample for hormone extraction (*Prohaska et al., 2013a*) and 19 g in the case of semen (*Gottfried & Chieffi, 1967*). In the majority of the studies, the obtention of samples in these quantities was met through the sacrifice of specimens, or through the interaction with captive elasmobranchs.

The sample mass required for the extraction of hormones in tissues like blood, gonad, histotroph and yolk usually entailed the use of a lethal methodology in 69.0% (*n* = 40) of the analyses (Fig. 2B). Most blood studies included sample collection carried out through cardiac, caudal or dorsal venepuncture (Table 2). Studies that did not required a lethal methodology (31.0%; *n* = 18) were considered invasive due to the capture or manipulation needed for the collection of samples. In addition to extraction kits, the main methods for hormones analyses in chondrichthyans were solvent/ether extractions using diethyl ether, ethyl acetate, and petroleum ether; although other substances such as chloroform: methanol, dichloromethane, ethyl acetate/cyclohexane, and benzene ether were also used (Table S1).

Some of the first papers that described steroid hormones in tissues other than blood were carried out via PC/TLC-GC by *Simpson, Wright & Gottfried (1963)* and *Simpson, Wright & Hunt (1964)* and *Gottfried & Chieffi (1967)* using semen of *Squalus acanthias* and *S. stellaris*, respectively. More than two decades later, analyses of histotroph, gonads and plasma from *S. canicula* and *T. marmorata* were performed via RIA (*Fasano et al., 1992*; *Garnier, Sourdaine & Jégou, 1999*). Analyses of steroid hormones from yolk were only studied in *S. tiburo* by RIA (*Manire et al., 2004*). The use of skeletal muscle was first

analysed by *Barnett et al. (2009)* and further by *Prohaska et al. (2013a)*, both via RIA. The latter suggested the use of muscle as an alternative for the study of protected elasmobranchs, since a correlation between steroid concentrations in plasma and muscle was observed (*Hammerschlag & Sulikowski, 2011*; *Prohaska et al., 2013a*).

All the analysed steroids used for reproductive assessment according to each biological matrix are detailed in Table 2. The average number of steroid hormones analysed per study that used plasma was 3.3 (S.D. ± 1.7); 4.2 (S.D. ± 1.2) for serum; 5.3 (S.D. ± 3.2) for gonad; 1.3 (S.D. ± 0.6) for semen; 3.0 for muscle; and 7.0 (S.D. ± 4.2) for histotroph. To date, most of the studies have focused on the analysis of several tissues using immunoassay-based methods such as RIA or ELISA/EIA and therefore the antigen-antibody complex interaction (Fig. 2A). Overall, the results from these analyses of the six biological matrices in several species are the current basis of the endocrinological knowledge of chondrichthyans in terms of reproductive biology. This scientific effort has allowed the detection and quantification of a wide diversity of steroid hormones and furthered the knowledge of their role in the reproduction of these cartilaginous fishes.

## Steroid hormones and reproductive biology

Since 1963, a total of 19 steroid hormones have been detected in six biological matrices from 34 species of chondrichthyans (Table 3). The analysed steroids comprise six androgens including *T*, 5α-dihydrotestosterone (DHT), 11-ketotestosterone (11-KT), androstenedione (A4), 11-ketoandrostenedione (11KA), and 5α-androstane-3α,17β-diol (3α-diol); four progestogens: P4, 17-hydroxyprogesterone (17-OHP), dihydroprogesterone (DHP), and 17-hydroxypregnenolone (17P5); three oestrogens: oestrone (E1), E2, as well as oestriol (E3); four glucocorticoids: corticosterone (CORT), cortisone (*E*), cortisol (*F*), and 11-deoxycortisol (*S*); and 11-deoxycorticosterone (DOC) and 11-dehydrocorticosterone (11-DHC) as the only two mineralocorticoids (*Idler, 2012*; *Melmed et al., 2015*; Fig. 4).

Ten of these 19 steroids were quantified in blood and nine in gonads, including the detection of 3α-diol solely in the testes of *S. canicula* (*Fasano et al., 1992*; *Garnier, Sourdaine & Jégou, 1999*). The use of LC-MS/MS allowed the simultaneous detection of 13 of such steroids in both plasma and histotroph (*Lyons & Wynne-Edwards, 2019*; Table 2). Other matrices such as semen allowed the detection of steroid hormones not found in blood, gonads or histotroph, such as 17P5 (*Gottfried & Chieffi, 1967*). In recent RIA studies, the use of skeletal muscle has allowed the detection of P4, *T* and E2 in the sharks *S. acanthias* and *R. terranovae* (*Prohaska et al., 2013a*).

According to the reviewed literature, a brief summary of the functions of steroid hormones in the reproduction of chondrichthyans, biological matrices and methods is provided below. These descriptions constitute a general summary of the observations published to date; therefore, specific functions of each steroid should be considered for each species, as differences between reproductive strategies, sexes and similar taxa have been reported.

Table 3 Steroid hormones analysed for the study of chondrichthyan reproduction during 1963–2020.

| | Hormones | Type | Analysed species | Number species |
|---|---|---|---|---|
| 1 | 17β-oestradiol (E2) | Oestrogen | All species in Table 1 | 34 |
| 2 | Testosterone (T) | Androgen | All species in Table 1 | 34 |
| 3 | Progesterone (P4) | Progestogen | All species in Table 1; except *H. colliei, S. stellaris* | 33 |
| 4 | 5α-dihydrotestosterone (DHT) | Androgen | *R. typus, C. taurus, C. leucas, C. plumbeus, N. brevirostris, S. canicula, S. tiburo, R. eglanteria, T. marmorata, H. sabinus, M. alfredi* | 11 |
| 5 | 11-ketotestosterone (11-KT) | Androgen | *H. colliei, C. laticeps, S. canicula, S. tiburo, U. halleri* | 7 |
| 6 | Corticosterone (CORT) | Glucocorticoid | *N. brevirostris, S. tiburo, T. marmorata, H. sabinus, U. halleri* | 7 |
| 7 | Estrone (E1) | Oestrogen | *S. canicula, T. marmorata, H. americanus, U. halleri* | 4 |
| 8 | Androstenedione (A4) | Androgen | *S. canicula, R. bonasus, U. halleri* | 3 |
| 9 | Estriol (E3) | Oestrogen | *T. marmorata, U. halleri* | 2 |
| 10 | 17-hydroxyprogesterone (17-OHP) | Progestogen | *S. canicula, U. halleri* | 2 |
| 11 | 11-deoxycorticosterone (DOC) | Mineralocorticoid | *S. acanthias, T. marmorata* | 2 |
| 12 | Cortisol (F) | Glucocorticoid | *T. marmorata, U. halleri* | 2 |
| 13 | Cortisone (E) | Glucocorticoid | *T. marmorata, U. halleri* | 2 |
| 14 | 11-deoxycortisol (S) | Glucocorticoid | *U. halleri* | 1 |
| 15 | Dihydroprogesterone (DHP) | Progestogen | *S. tiburo* | 1 |
| 16 | 17-hydroxypregnenolone (17P5) | Progestogen | *S. stellaris* | 1 |
| 17 | 11-ketoandrostenedione (11KA4) | Androgen | *S. tiburo* | 1 |
| 18 | 5α-androstane-3α,17β-diol (3α-diol) | Androgen | *S. canicula* | 1 |
| 19 | 11-dehydrocorticosterone (11-DHC) | Mineralocorticoid | *U. halleri* | 1 |

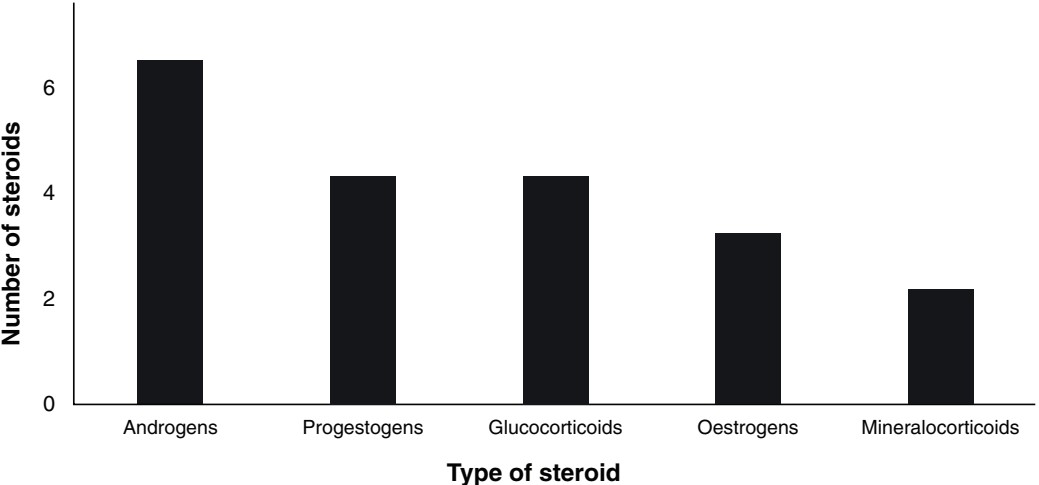

Figure 4 Number of analysed steroids according to their type (*n* = 19).

## Testosterone

High concentrations of this androgen have been related to follicle development and to the maintenance of spermatozoa along with E2 in females (*Manire, Rasmussen & Gross, 1999*;

*Tricas, Maruska & Rasmussen, 2000*). Additionally, an increase of *T* was observed during the mating seasons of several elasmobranchs, suggesting that this steroid could play an important role in their reproductive ethology (*Rasmussen & Gruber, 1993*; *Manire et al., 1995*; *Garnier, Sourdaine & Jégou, 1999*; *Mull, Lowe & Young, 2010*). In males, several studies described an increase of this androgen during spermatogenesis and sperm transportation, which could indicate a role in the maturation and motility of gametes (*Heupel, Whittier & Bennett, 1999*; *Sulikowski, Tsang & Howell, 2005*). Additionally, *T* could be an anabolic agent for the growth of testes, reproductive ducts and claspers (*Sulikowski, Tsang & Howell, 2005*; *Lyons & Wynne-Edwards, 2019*). In contrast to other vertebrates, knowledge regarding the functions of *T* in the body or in the development of gonads is still scarce (*Awruch, 2013*). This androgen has been quantified in six biological matrices using the five analytical methods described in this paper. A total of 52 papers have included *T* in their analyses, which constitutes 88.1% of the reviewed literature (Table 2).

### 5α-dihydrotestosterone

In chondrichthyans, DHT has been suggested as an important steroid for spermatogenesis along with *T* as a possible precursor of E2 (*Manire & Rasmussen, 1997*; *Tricas, Maruska & Rasmussen, 2000*; *Awruch, 2013*). However, there is no evidence regarding an aromatisation of DHT into oestrogens such as E2, so this hypothesis is unlikely. In terms of steroid synthesis control, *Tricas, Maruska & Rasmussen (2000)* suggested that DHT could provide an inhibitory feedback in the production of gonadotropin-releasing hormone (GnRH), based on their observations in the ray *Hypanus sabinus*. Additionally, an increase of DHT and *T* was observed during ovulation and later during histotroph nourishment. This could suggest that the action of DHT is related to the sexual differentiation in batoids as observed in other vertebrates (*Tricas, Maruska & Rasmussen, 2000*; *Lyons & Wynne-Edwards, 2019*). In viviparous sharks such as *S. tiburo*, DHT was related to reproductive behaviour, as the highest concentration of this steroid occurred during preovulation and mating stages (*Manire et al., 1995*). DHT was detected and quantified in blood, gonads and histotroph of elasmobranchs using RIA and ELISA (Table 2). There are no detections of DHT in holocephans to date. From the reviewed literature, a total of 13 papers (22.0%) have included this steroid in their analyses (Table 2).

### 11-ketotestosterone

In female elasmobranchs, 11-KT may play a role in the synthesis of oestrogens and ovarian follicle development, although most of the research has been focused on males (*Manire, Rasmussen & Gross, 1999*). In this regard, a higher concentration of 11-KT could favour the sexual development of juvenile males, with an increase during the peak of sperm storage influencing sperm maturation and preparation for mating (*Garnier, Sourdaine & Jégou, 1999*; *Mull, Lowe & Young, 2008*). This androgen showed a negative correlation with photoperiod in specimens of *U. halleri* with levels of 11-KT decreasing with daylight. The effect of light on the synthesis of some steroid hormones such as 11-KT could explain

differences in sexual segregation regarding depth and habitat use for males, although research in this topic remains scarce. In holocephans, levels of 11-KT were 300% higher during pre-parturition stages, which suggest that this steroid could be related not only to parturition, but to sexual behaviour favouring the next reproductive cycle (*Barnett et al., 2009*). In general, it is likely that 11-KT is involved in the sexual maturity and mating behaviour of male chondrichthyans, although more research is needed in order to understand the exact role in their development and further reproduction. 11-KT has been detected and quantified in blood, gonads and muscle of chondrichthyans using RIA (Table 2). From the reviewed literature, a total of six papers (10.2%) have included 11-KT in their analyses (Table 2).

### Androstenedione

In batoids, A4 was higher during the greater fertile period and early gestation of *Rhinoptera bonasus*, while there were no differences in A4 along with E2 in the analysed males (*Sheldon et al., 2018*). A similar increase was observed in early and mid-term females just before the appearance of claspers in embryos of the round stingray *U. halleri*, which could suggest an intermediary role for sex differentiation along with 17-OHP (*Lyons & Wynne-Edwards, 2019*). In contrast, the concentrations detected in *S. canicula* demonstrated an increase before a peak in sperm reserves, and A4 was consistently detected in the testis as one of the principal steroids along with *T* and P4 (*Garnier, Sourdaine & Jégou, 1999*). A4 has been quantified in plasma and testis of *S. canicula* and in the plasma of both sexes of *R. bonasus* using RIA, as well as in the histotroph of *U. halleri* (Table 2). Three papers (5.1%) included A4 in their analyses (Table 2).

### 11-ketoandrostenedione

This steroid was detected in serum of *S. tiburo* using RIA and appears only in the study of *Manire, Rasmussen & Gross (1999)*. It is possible that an increase in 11KA4 plays a role in sperm storage and in the maintenance of pregnancy in viviparous species, as the concentrations rise from mating until prior to parturition. In males, the function of 11KA4 is unclear, as levels appear to be similar throughout the annual study periods in mature individuals. Further research involving different maturity stages could provide the first insights regarding the role of 11KA4 in most chondrichthyans. The paper of *Manire, Rasmussen & Gross (1999)* constitutes 1.7% of the reviewed literature.

### 5α-androstane-3α,17β-diol

This hormone was quantified throughout the year in plasma and testis of *S. canicula* with significantly higher concentrations during winter, just prior to an increase in steroid hormones such as E2, E1, DHT,11-KT and a peak in sperm reserves (*Garnier, Sourdaine & Jégou, 1999*). While 3α-diol may play a role in the preparation of spermatogenesis, the metabolism of this steroid and its effects on the reproduction of chondrichthyans remain unclear. The paper of *Garnier, Sourdaine & Jégou (1999)* used RIA as the analytical method and constitutes 1.7% of the reviewed literature.

### Progesterone

In females, this progestogen was suggested to act as an antagonist of E2 and consequently decrease the synthesis of vitellogenin in the liver (*Tsang & Callard, 1987*; *Tricas, Maruska & Rasmussen, 2000*; *Prisco et al., 2008*; *Mull, Lowe & Young, 2010*). An increase of P4 along with 11KA4 could be related to the beginning and maintenance of pregnancy in viviparous species since the concentration of these hormones drop prior to parturition (*Manire, Rasmussen & Gross, 1999*; *Mull, Lowe & Young, 2010*). In contrast, it is possible that a decrease in P4 is related to the regulation of encapsulation and oviposition since low levels of P4 were detected after ovulation in oviparous species (*Koob, Tsang & Callard, 1986*; *Sulikowski, Tsang & Howell, 2004*). The function of P4 in males remains unclear, with different studies showing contrasting results (*Awruch, 2013*). However, it is probable that P4 is linked to the synthesis of *T* and other steroid hormones, which could partially be related to sexual maturation (*Simpson, Wright & Gottfried, 1963*; *Rasmussen & Gruber, 1993*). P4 has been quantified in five biological matrices using the five analytical methods described in this paper. A total of 44 papers have included P4 in their analyses, which constitutes 74.6% of the reviewed literature (Table 2).

### 17-hydroxyprogesterone

This hormone was detected throughout the complete reproductive cycle of *S. canicula* with no significant differences among the analysed months (*Garnier, Sourdaine & Jégou, 1999*). In contrast, hormones such as A4 and 17-OHP could be related to sex differentiation in male embryos of *U. halleri* (*Lyons & Wynne-Edwards, 2019*). This progestogen has been quantified in plasma and testis of *S. canicula* and in the plasma and histotroph of *U. halleri* using RIA and LC-MS/MS, respectively (Table 2). Two papers (3.4%) included 17-OHP in their analyses (Table 2).

### Dihydroprogesterone

The study of *Manire, Rasmussen & Gross (1999)* demonstrated that levels of DHP significantly increased during maturation in females of *S. tiburo*, but decreased before spermatogenesis in adult males along with an increase in E2 and 11-KT levels. This could be related to the synthesis of E2 and 11-KT, although this process remains unclear. The analysis of DHP was made in serum of this carcharhinid using RIA. The paper of *Manire, Rasmussen & Gross (1999)* constitutes 1.7% of the reviewed literature.

### 17-hydroxypregnenolone

The study of *Gottfried & Chieffi (1967)* was the only one to include 17P5 by analysing semen of *S. stellaris* using TLC-GC. However, this study was purely technical, focusing on hormone detection without any biological inferences. 17P5 may have receptors in the brain of elasmobranchs or may be involved in the synthesis of other hormones such as 17-OHP (*Diotel et al., 2011*). However, there is no evidence to date to support the functions of 17P5 in chondrichthyans. The paper of *Gottfried & Chieffi (1967)* constitutes 1.7% of the reviewed literature.

### Estrone

This steroid was at higher levels before the gestation period of *T. marmorata*, with no detections observed during the pregnancy of this species (*Di Prisco, Vellano & Chieffi, 1967*). In contrast, levels of E1 and androstenedione (A4) were only detected in the histotroph of early and mid-term pregnant females of *U. halleri* (*Lyons & Wynne-Edwards, 2019*). This could suggest a transfer of energy and substances for the development of these species; however, evidence about these mechanisms remains scarce. This oestrogen has been quantified in plasma and histotroph of *U. halleri*, and in the plasma of *T. marmorata* using LC-MS/MS and TLC-GC, respectively (Table 2), but has not been investigated in holocephans. From the reviewed literature, four papers (6.8%) have included E1 in their analyses (Table 2).

### 17β-oestradiol

In chondrichthyan females, E2 has been linked to the synthesis of vitellogenin in the liver (*Prisco et al., 2008*), follicle development in the ovary (*Koob, Tsang & Callard, 1986*; *Heupel, Whittier & Bennett, 1999*; *Manire, Rasmussen & Gross, 1999*; *Henningsen et al., 2008*), development and functions of the oviducal gland including the storage of spermatozoa along with T (*Koob, Tsang & Callard, 1986*; *Tsang & Callard, 1987*; *Manire et al., 1995*; *Tricas, Maruska & Rasmussen, 2000*; *Gelsleichter & Evans, 2012*; *Awruch, 2013*), and the synthesis of protein for the capsules of oviparous species (*Dodd & Goddard, 1961*). An increase of E2 has been related to the secretion of histotroph in the uterus of batoids (*Snelson et al., 1997*; *Tricas, Maruska & Rasmussen, 2000*; *Lyons & Wynne-Edwards, 2019*), as well as to the formation of the placental connection in some viviparous sharks (*Manire et al., 1995*). It has been proposed that E2 is linked to the presence of relaxin and therefore involved in the preparation for parturition (*Tsang & Callard, 1987*; *Koob, Laffan & Callard, 1984*). In recent years, it was suggested that E2 and T could play an important role in sexual differentiation of batoids such as *U. halleri* (*Lyons & Wynne-Edwards, 2019*). In males, an increase of E2 was related to spermatogenesis (*Tricas, Maruska & Rasmussen, 2000*; *Sulikowski, Tsang & Howell, 2004*) and the development of gonads along with other steroids such as P4, T and DHT (*Gelsleichter et al., 2002*). This oestrogen has been quantified in six biological matrices using the analytical methods described in this review except for PC/TLC-GC. From the reviewed literature, a total of 49 papers (83.0%) have included E2 in their analyses (Table 2).

### Estriol

In female batoids, E3 was more abundant in immature individuals compared to mature individuals of *T. marmorata* (*Di Prisco, Vellano & Chieffi, 1967*). In a recent study, this steroid was postulated to influence sex differentiation along with E2, as it was detected in the early development of female-only litters of *U. halleri* (*Lyons & Wynne-Edwards, 2019*). This oestrogen was quantified in the plasma and histotroph of *U. halleri* and in plasma of *T. marmorata* using LC-MS/MS and TLC-GC, respectively. From the reviewed literature, only two papers (3.4%) have included E3 in their analyses (Table 2).

### Cortisone

This glucocorticoid was analysed by *Di Prisco, Vellano & Chieffi (1967)* using plasma of *T. marmorata* and TLC-GC. The concentrations of *E* where consistently lower than *F*, with a decrease of these steroids from immaturity to gestation in the analysed females. In contrast, the concentrations of CORT increased. The study of *Lyons & Wynne-Edwards (2019)* did detect the hormone, yet it indicated concentrations below the limit of quantification in plasma, along with a lack of detections in the histotroph of *U. halleri* (Table 2). Two papers (3.4%) using LC-MS/MS and TLC-GC included *E* in their reproductive analyses (Table 2).

### Corticosterone

An increase in CORT levels was linked to sexual development and mating in males of elasmobranch species such as *Negaprion brevirostris*, *S. tiburo* and *H. sabinus* (*Rasmussen & Gruber, 1993*; *Manire et al., 2007*). In females of *S. tiburo* and *T. marmorata*, CORT was detected during mating, sperm storage and early gestation (*Di Prisco, Vellano & Chieffi, 1967*; *Snelson et al., 1997*; *Manire et al., 2007*). However, this pattern was not observed in females of *H. sabinus* where CORT levels were low during the entire study period (*Snelson et al., 1997*), but were elevated in late gestation and parturition in the study of *Manire et al. (2007)*. Although CORT was detected in the plasma and histotroph of *U. halleri*, the concentrations were below the limit of quantification, so no biological inferences were provided (*Lyons & Wynne-Edwards, 2019*). CORT has been quantified in blood and histotroph of several elasmobranch species using RIA, TLC-GC or LC-MS/MS, but it has not been investigated in holocephans. From the reviewed literature, five papers (8.5%) have included CORT in their analyses (Table 2).

### Cortisol

In terms of the reproductive biology of batoids, *Lyons & Wynne-Edwards (2019)* observed that *F* was only present in post-ovulatory females and one early term female of *U. halleri*. In another study by *Di Prisco, Vellano & Chieffi (1967)*, *F* was detected during the whole reproductive cycle of *T. marmorata* in concentrations that were permanently higher than *E*. This glucocorticoid has been analysed in the plasma of *T. marmorata* and *U. halleri* using TLC-GC and LC-MS/MS, respectively, since the concentrations of these hormones were below the limit of quantification in the histotroph of *U. halleri*. Two papers (3.4%) included *F* in their analyses (Table 2).

### 11-deoxycortisol

Although *S* was recently detected during pregnancy in plasma and histotroph of *U. halleri*, the observed concentrations were below the limit of quantification, hence limiting our ability to draw conclusions regarding its function. The paper of *Lyons & Wynne-Edwards (2019)* using LC-MS/MS constitutes 1.7% of the reviewed literature.

### 11-Deoxycorticosterone

In elasmobranchs, DOC was detected during different maturity stages in *T. marmorata* females, which could indicate its role in the synthesis of CORT and the regulation of

energy and stress (*Di Prisco, Vellano & Chieffi, 1967*). In the case of male sharks, DOC was suggested as a protein-bound component of seminal plasma in *S. acanthias*, yet its effect on sperm motility or other influence on semen remains unknown (*Simpson, Wright & Hunt, 1964*). DOC has been detected in semen of *S. acanthias* and plasma of *T. marmorata* using PC and TLC-GC, respectively (Table 2). Three papers included DOC in their analysis, which constitutes 5.1% of the reviewed literature.

### 11-dehydrocorticosterone

In chondrichthyans, 11-DHC was first detected in late term females of the round stingray *U. halleri* by *Lyons & Wynne-Edwards (2019)*. However, the concentrations were found below the limit of quantification. It is possible that the presence of 11-DHC could be related to the conversion of this steroid to *F* or CORT by enzymes such as 11β-hydroxysteroid dehydrogenase (*Pelletier, 2010*). Nevertheless, the scientific research related to this topic is scarce and there is no evidence of its effects on chondrichthyan metabolism and reproduction. 11-DHC was detected using LC-MS/MS in the paper of *Lyons & Wynne-Edwards (2019)* which constitutes 1.7% of the reviewed literature.

General models illustrating the function of steroid hormones in male and female chondrichthyans are provided in Figs. 5 and 6. These theoretical diagrams summarise the observations made to date in all the analysed species and can highly differ between taxa. In this regard, considerations for each reproductive strategy, embryonic nutrition and species should be cautiously considered since research in most of the hormones is currently insufficient. A brief summary of the effects of steroid hormones during each biological phenomenon are mentioned below. Further details are also reviewed in *Maruska & Gelsleichter (2011)*, *Gelsleichter & Evans (2012)* and *Awruch (2013)*.

Overall, male chondrichthyan steroidogenesis is initiated in the testis by the action of the luteinizing hormone (LH) and follicle-stimulating hormone (FSH) produced in the pituitary gland on the Sertoli and Leydig cells (Fig. 5; *Callard et al., 1989*). During sexual maturation, some steroids such as *T*, P4, 11-KT and CORT play a role as anabolic agents promoting the mitosis and further development of testis, reproductive ducts, seminal glands and claspers (*Rasmussen & Gruber, 1993*; *Manire, Rasmussen & Gross, 1999*; *Sulikowski, Tsang & Howell, 2005*; *Manire et al., 2007*; *Lyons & Wynne-Edwards, 2019*). In mature individuals, an increase of *T*, DHT and oestrogens such as E2 will favour spermatogenesis, while the particular function of androgens such as *T* are related to sperm motility and transportation throughout the paired reproductive ducts (*Manire & Rasmussen, 1997*; *Heupel, Whittier & Bennett, 1999*; *Tricas, Maruska & Rasmussen, 2000*; *Sulikowski, Tsang & Howell, 2004*, *2005*). The synthesis of seminal plasma will involve the presence of DOC as a component of this substance (*Simpson, Wright & Hunt, 1964*), where A4 and 3α-diol occur during the peak of sperm reserves prior the reproductive season (*Garnier, Sourdaine & Jégou, 1999*). The presence of A4 and 3α-diol could be a precursor for an increase in *T*, DHT and CORT that, in turn, will promote mating behaviours and the further copulation with females (Fig. 5; *Rasmussen & Gruber, 1993*; *Manire et al., 1995*; *Garnier, Sourdaine & Jégou, 1999*; *Manire et al., 2007*; *Mull, Lowe & Young, 2010*).

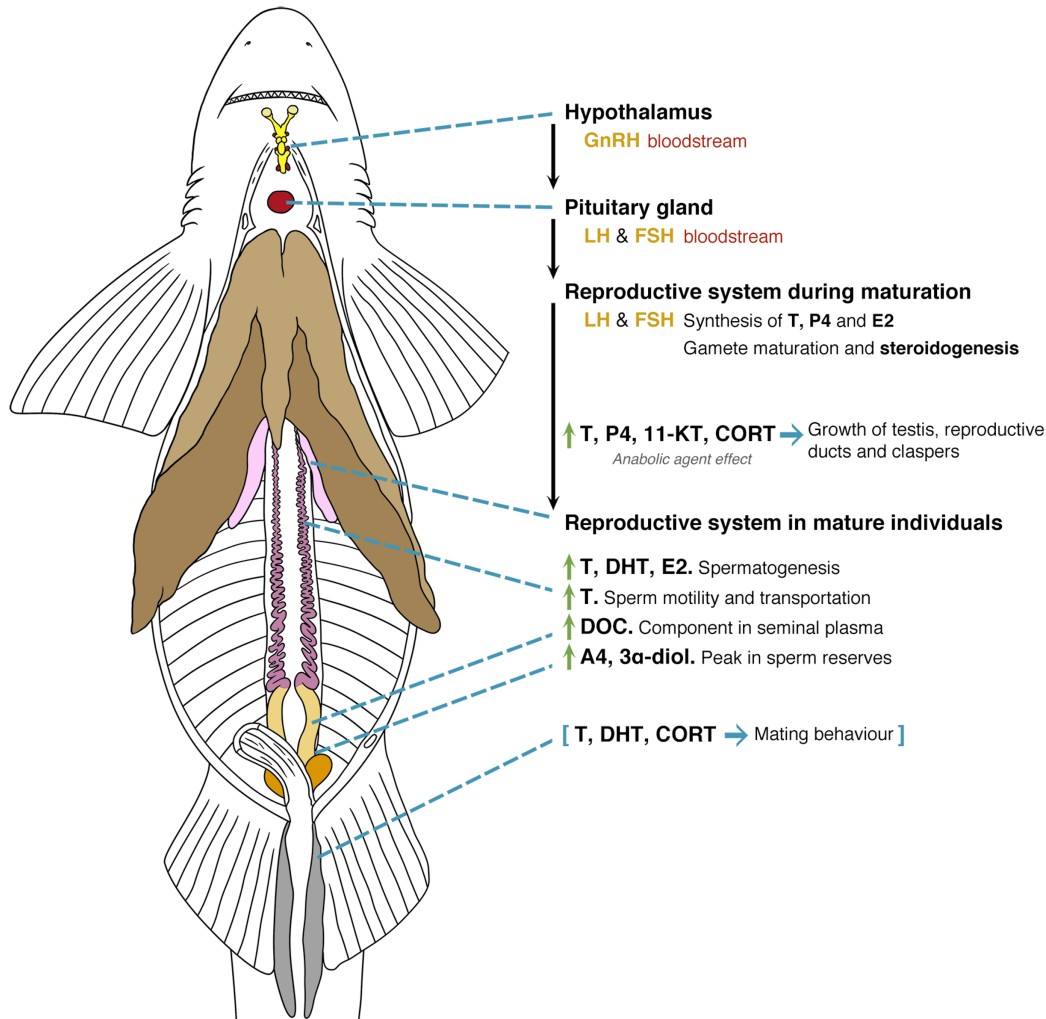

**Hypothalamus**
**GnRH** bloodstream

**Pituitary gland**
**LH** & **FSH** bloodstream

**Reproductive system during maturation**
**LH** & **FSH** Synthesis of **T, P4** and **E2**
Gamete maturation and **steroidogenesis**

↑**T, P4, 11-KT, CORT** → Growth of testis, reproductive
*Anabolic agent effect* ducts and claspers

**Reproductive system in mature individuals**

↑**T, DHT, E2.** Spermatogenesis
↑**T.** Sperm motility and transportation
↑**DOC.** Component in seminal plasma
↑**A4, 3α-diol.** Peak in sperm reserves

[ **T, DHT, CORT** → Mating behaviour ]

**Figure 5 General model of the endocrinology of male chondrichthyans in terms of steroid hormones and reproductive biology.** Abbreviations: GnRH, Gonadotropin releasing hormone; LH, luteinizing hormone; FSH, follicle stimulating hormone; E2, 17β-oestradiol; *T*, testosterone; P4, progesterone; DHT, 5α-dihydrotestosterone; 11-KT, 11-ketotestosterone; CORT, corticosterone; A4, androstenedione; DOC, 11 deoxycorticosterone; 3α-diol, 5α-androstane-3α,17β-diol.

Sexual maturation and reproductive mechanisms in mature females are influenced by the presence of LH and FSH (Fig. 6; *Maruska & Gelsleichter, 2011*; *Awruch, 2013*). This promotes the synthesis of P4, *T* and E2 at the beginning of ovarian follicle development by its production in the granulosa and theca cells (*Gelsleichter & Evans, 2012*). During sexual maturation and the beginning of each reproductive cycle in mature specimens, the effect of E2 on the liver favours the synthesis of vitellogenin, which is essential for oocyte development (*Prisco et al., 2008*). During this process, the effects of DOC and 11-DHC are linked to the synthesis of CORT and the regulation of energy and stress (*Di Prisco, Vellano & Chieffi, 1967*; *Pelletier, 2010*; *Lyons & Wynne-Edwards, 2019*). In this regard, it is likely that specific steroid hormones such as

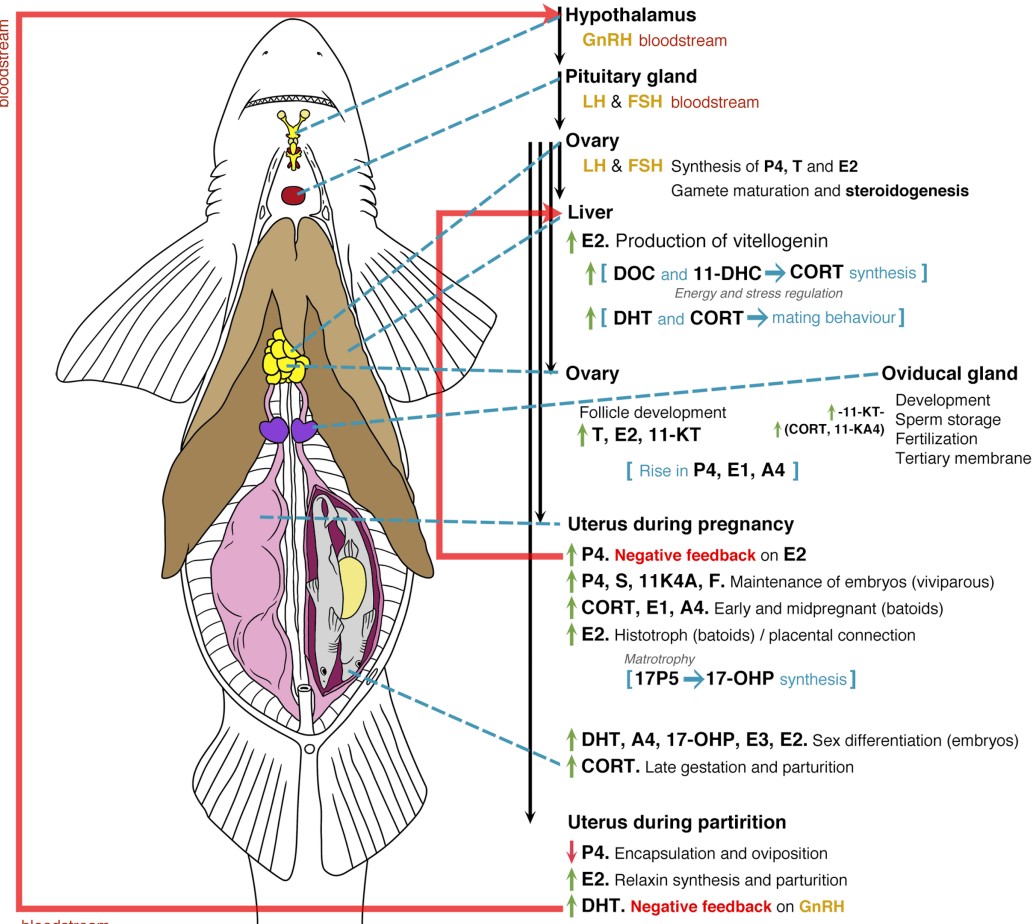

**Figure 6 General model of the endocrinology of female chondrichthyans in terms of steroid hormones and reproductive biology.** Abbreviations: GnRH, Gonadotropin releasing hormone; LH, luteinizing hormone; FSH, follicle stimulating hormone; E2, 17β-oestradiol; $T$, testosterone; progesterone (P4); DHT, 5α-dihydrotestosterone; 11-KT, 11-ketotestosterone; CORT, corticosterone; A4, androstenedione; 17-OHP, 17-hydroxyprogesterone; E1, estrone; E3, estriol; $E$, cortisone; $F$, cortisol; $S$, 11-deoxycortisol; DHP, dihydroprogesterone; DOC, 11 deoxycorticosterone; 11KA4, 11-ketoandrostenedione; 3α-diol, 5α-androstane-3α,17β-diol; 17P5, 17 hydroxypregnenolone; 11-DHC, 11-dehydrocorticosterone.

1α-hydroxycorticosterone play an important role during all reproductive phenomena; however, their functions in reproductive processes are still unclear (*Iki et al., 2020*). Follicle development in preparation for the beginning of the mating seasons or reproductive cycles is favoured by an increase in *T*, E2 and 11-KT, while the development of reproductive ducts and the oviducal gland will be related to such steroids and to the action of 11-KT (*Koob, Tsang & Callard, 1986*; *Heupel, Whittier & Bennett, 1999*; *Manire, Rasmussen & Gross, 1999*; *Tricas, Maruska & Rasmussen, 2000*; *Henningsen et al., 2008*).

After mating, the action of CORT and 11-KA4 allows sperm storage in preparation for conception and synthesis of a tertiary membrane. This structure becomes the egg case in oviparous species or a protective membrane in viviparous embryos (*Hamlett, 2011*).

A rise in P4, E1 and A4 occurs during the preparation for the passing of fertilised oocytes from the oviducal gland to the uterus (*Di Prisco, Vellano & Chieffi, 1967*; *Koob, Tsang & Callard, 1986*; *Sulikowski, Tsang & Howell, 2004*; *Sheldon et al., 2018*). During pregnancy, this increase in P4 acts as a negative feedback for the synthesis of E2, resulting in a reduction of the vitellogenin synthesis (*Tsang & Callard, 1987*; *Tricas, Maruska & Rasmussen, 2000*; *Prisco et al., 2008*; *Mull, Lowe & Young, 2010*). As previously mentioned, all these effects would change according to the species. For instance, species showing oophagy or adelphophagy could present different routes since unfertilised or fertilised oocytes are provided for the nutrition of embryos (*Compagno, Dando & Fowler, 2005*; *Hamlett, 2011*). Bloodstream levels of P4, S and 11KA4 favour the conditions for the maintenance of embryos, with the presence of CORT, E1 and A4 playing a role in the potential regulation of energy and stress (Fig. 6; *Di Prisco, Vellano & Chieffi, 1967*; *Manire, Rasmussen & Gross, 1999*; *Sheldon et al., 2018*; *Lyons & Wynne-Edwards, 2019*).

In the case of histotrophic batoids, a rise in E2 affects the ciliated cells in the uterus for the synthesis of the histotroph (*Snelson et al., 1997*; *Tricas, Maruska & Rasmussen, 2000*; *Lyons & Wynne-Edwards, 2019*), while in placental species this E2 rise is linked to the consumption of the yolk and the start of a placental connection (*Manire et al., 1995*). Specific steroids involving androgens and oestrogens such as DHT, A4, 17-OHP, E3 and E2 will play a role in the sex differentiation of embryos (*Tricas, Maruska & Rasmussen, 2000*; *Lyons & Wynne-Edwards, 2019*), with high levels of CORT present during the late gestation and prior to parturition (*Manire et al., 2007*). In the last phases of the reproductive cycle, a drop in P4 will affect the original conditions for the maintenance of embryos or the start of oviposition (*Koob, Tsang & Callard, 1986*; *Manire, Rasmussen & Gross, 1999*; *Sulikowski, Tsang & Howell, 2004*; *Mull, Lowe & Young, 2010*). The absence of P4 allows the recommencement of E2 synthesis, which plays a role in the production of relaxin for parturition (*Koob, Laffan & Callard, 1984*; *Tsang & Callard, 1987*). The oviposition or the birth of the pups likely stimulates an increase in DHT levels that could act as a negative feedback for the synthesis of gonadotropin-releasing hormone (Fig. 6; *Tricas, Maruska & Rasmussen, 2000*).

## Scientific research

Current knowledge of the reproductive endocrinology in chondrichthyans is primarily based on the analysis of E2, T and P4, as these are the three main sex steroids in vertebrates (*Maruska & Gelsleichter, 2011*; *Idler, 2012*; *Awruch, 2013*). Due to their importance in reproduction, these hormones have been described in many of the studied species (Table 3). However, the detection, quantification and analysis of other steroid hormones is considerably limited in chondrichthyan literature. For instance, ninety three percent ($n = 55$) of the reviewed papers have described less than six steroids, and only four papers (6.8%) detected 7–13 steroids (Fig. 2C). The lack of a reliable characterisation of the chondrichthyan reproductive cycle through hormonal profiles is likely related to the limited number of studies reporting on these other steroid hormones (*Anderson et al., 2018*).

The analysis of E2, *T* and P4 can provide substantial information regarding reproduction. However, the effect of potent androgens such as DHT, oestrogens like E1 or E3, as well as relevant glucocorticoids such as 1α-hydroxycorticosterone would be greatly beneficial for the characterisation of the reproductive cycle in both male and female chondrichthyans (*Rasmussen & Gruber, 1993*; *Maruska & Gelsleichter, 2011*; *Iki et al., 2020*). As mentioned by several authors such as *Anderson et al. (2018)*, there is currently no reliable hormone indicators for the identification of relevant aspects such as maturity, mating seasons, gestation, partum or postpartum in chondrichthyans. In this regard, it is expected that the inclusion of more hormones, whether in sequential or simultaneous analyses, could lead to a better description and greater understanding of reproduction, as some steroids such as 11-KT, A4 or 17-OHP could provide insights about specific conditions in the reproductive cycle of chondrichthyans (*Garnier, Sourdaine & Jégou, 1999*; *Mull, Lowe & Young, 2008*; *Lyons & Wynne-Edwards, 2019*).

Scientific research of endocrinology in chondrichthyans has increased since the 1990s (Fig. 2D). For instance, only seven articles were published between 1961 and 1990, constituting an average of 0.2 papers per year compared with the 1.8 papers published per year after 1990. This increase is linked to improvements in technological and analytical methodologies, as well as the rising interest for the management of these species around the world (*Hammerschlag & Sulikowski, 2011*; *Cisneros-Montemayor et al., 2020*).

During 1963–2020, the species of sharks with the greatest number of hormones analysed were *S. tiburo* and *S. canicula*, with eight and 13 steroids, respectively. In the case of batoids, a total of 10 and 13 steroids were detected for *T. marmorata* and *U. halleri*, respectively (Table 4). The spotted ratfish *H. colliei* is the only chimaera species in which hormones were analysed, namely E2, *T* and 11-KT (*Barnett et al., 2009*), although additional studies exist related to nuclear receptors that were not considered in this review (*Katsu et al., 2010*, *2019*).

The scientific research on steroid hormones and their functions in the reproduction of chondrichthyans is mainly based on a selected number of species used as biological models. Eighty six percent of the scientific effort has been carried out in the northern hemisphere ($n = 51$ papers), including developed countries like the USA, Canada, England, Italy and Japan (Fig. 7). This is highly related to high quality facilities, available research instrumentation and supplies, as well as financial support for the development of science, all of which are not frequently provided in developing or non-developed countries. The rest of the scientific effort (13.8%; $n = 8$ papers) originated from countries in the southern hemisphere such as Australia and Argentina. The latter is the only country that has explored steroid hormones in the reproduction of elasmobranchs in South America. Regrettably, research is noticeably lacking in areas with a high biodiversity of chondrichthyans such as the Caribbean and Red Sea, the Eastern Tropical Pacific, the Indo Pacific Ocean, the polar seas and nearly the entire coast of Africa (Fig. 7).

Considering the ecological richness of chondrichthyans, the analysis of additional species from different habitats could lead to a better understanding of the function of steroid hormones on the biology of sharks, rays and chimaeras. This would in turn benefit the reproductive assessment of commercial and protected species. The existence of

**Table 4 Steroid hormones analysed per chondrichthyan species during 1963–2020.**

| Species | E2 | T | P4 | DHT | 11-KT | CORT | E1 | A4 | E3 | 17-OHP | DOC | F | E | S | DHP | 17P5 | 11KA4 | 3α-diol | 11-DHC |
|---|---|---|---|---|---|---|---|---|---|---|---|---|---|---|---|---|---|---|---|
| *Hydrolagus colliei* | ● | ● | – | – | ● | – | – | – | – | – | – | – | – | – | – | – | – | – | – |
| *Notorynchus cepedianus* | ● | ● | ● | – | – | – | – | – | – | – | – | – | – | – | – | – | – | – | – |
| *Centroscymnus coelolepis* | ● | ● | ● | – | – | – | – | – | – | – | – | – | – | – | – | – | – | – | – |
| *Squalus acanthias* | ● | ● | – | – | – | – | – | – | – | – | ● | – | – | – | – | – | – | – | – |
| *Chiloscyllium plagiosum* | ● | ● | ● | – | – | – | – | – | – | – | – | – | – | – | – | – | – | – | – |
| *Hemiscyllium ocellatum* | ● | ● | ● | – | – | – | – | – | – | – | – | – | – | – | – | – | – | – | – |
| *Rhincodon typus* | ● | ● | ● | ● | – | – | – | – | – | – | – | – | – | – | – | – | – | – | – |
| *Stegostoma fasciatum* | ● | ● | ● | – | – | – | – | – | – | – | – | – | – | – | – | – | – | – | – |
| *Carcharodon carcharias* | ● | ● | ● | – | – | – | – | – | – | – | – | – | – | – | – | – | – | – | – |
| *Carcharias taurus* | ● | ● | ● | ● | – | – | – | – | – | – | – | – | – | – | – | – | – | – | – |
| *Carcharhinus leucas* | ● | ● | ● | ● | – | – | – | – | – | – | – | – | – | – | – | – | – | – | – |
| *Carcharhinus plumbeus* | ● | ● | ● | ● | – | – | – | – | – | – | – | – | – | – | – | – | – | – | – |
| *Galeocerdo cuvier* | ● | ● | ● | – | – | – | – | – | – | – | – | – | – | – | – | – | – | – | – |
| *Negaprion brevirostris* | ● | ● | ● | ● | – | ● | – | – | – | – | – | – | – | – | – | – | – | – | – |
| *Prionace glauca* | ● | – | ● | – | – | – | – | – | – | – | – | – | – | – | – | – | – | – | – |
| *Rhizoprionodon taylori* | ● | ● | ● | – | – | – | – | – | – | – | – | – | – | – | – | – | – | – | – |
| *Rhizoprionodon terranovae* | ● | ● | ● | – | – | – | – | – | – | – | – | – | – | – | – | – | – | – | – |
| *Cephalloscyllium laticeps* | ● | ● | ● | – | ● | – | – | – | – | – | – | – | – | – | – | – | – | – | – |
| *Scyliorhinus canicula* | ● | ● | ● | ● | ● | – | ● | ● | – | ● | – | – | – | – | – | – | – | ● | – |
| *Scyliorhinus stellaris* | ● | ● | – | – | – | – | – | – | – | – | – | – | – | – | – | ● | – | – | – |
| *Sphyrna tiburo* | ● | ● | ● | ● | ● | ● | – | – | – | – | – | – | – | – | ● | – | ● | – | – |
| *Mustelus schmitti* | ● | ● | ● | – | – | – | – | – | – | – | – | – | – | – | – | – | – | – | – |
| *Trygonorrhina dumerilii* | ● | ● | ● | – | – | – | – | – | – | – | – | – | – | – | – | – | – | – | – |
| *Amblyraja radiata* | ● | ● | ● | – | – | – | – | – | – | – | – | – | – | – | – | – | – | – | – |
| *Leucoraja erinacea* | ● | ● | – | – | – | – | – | – | – | – | – | – | – | – | – | – | – | – | – |
| *Leucoraja ocellata* | ● | ● | – | – | – | – | – | – | – | – | – | – | – | – | – | – | – | – | – |
| *Malacoraja senta* | ● | ● | ● | – | – | – | – | – | – | – | – | – | – | – | – | – | – | – | – |
| *Raja eglanteria* | ● | ● | ● | ● | – | – | – | – | – | – | – | – | – | – | – | – | – | – | – |
| *Torpedo marmorata* | ● | ● | ● | ● | – | ● | ● | – | ● | – | ● | ● | ● | – | – | – | – | – | – |
| *Hypanus sabinus* | ● | ● | ● | ● | – | ● | – | – | – | – | – | – | – | – | – | – | – | – | – |
| *Hypanus americanus* | ● | ● | – | – | – | ● | – | – | – | – | – | – | – | – | – | – | – | – | – |
| *Mobula alfredi* | ● | ● | ● | ● | – | – | – | – | – | – | – | – | – | – | – | – | – | – | – |
| *Rhinoptera bonasus* | ● | ● | – | – | – | – | – | ● | – | – | – | – | – | – | – | – | – | – | – |
| *Urobatis halleri* | ● | ● | ● | – | ● | ● | ● | ● | ● | ● | – | ● | ● | ● | – | – | – | – | ● |

**Note:**
E2, 17β-oestradiol; *T*, testosterone; P4, progesterone; DTH, 5α-dihydrotestosterone; 11-KT, 11-ketotestosterone; CORT, corticosterone; A4,androstenedione; 17-OHP, 17-hydroxyprogesterone; E1, estrone; E3, estriol; *E*, cortisone; *F*, cortisol; *S*, 11-deoxycortisol; DHP, dihydroprogesterone; DOC, 11 deoxycorticosterone; 11KA4, 11-ketoandrostenedione; 3α-diol, 5α-androstane-3α, 17b-diol; 17P5, 17-hydroxypregnenolone; 11-DHC, 11-dehydrocorticosterone. Black circles (●) indicate the steroid hormones that have been analysed according to each species; while dash (–) indicates the lack of analyses.

chondrichthyans in marine protected or non-protected areas should motivate the development of non-lethal methods for the analysis of reproduction through hormonal profiles, which could complement the study of their biology (*Hammerschlag & Sulikowski, 2011*; *Prohaska et al., 2013a*).

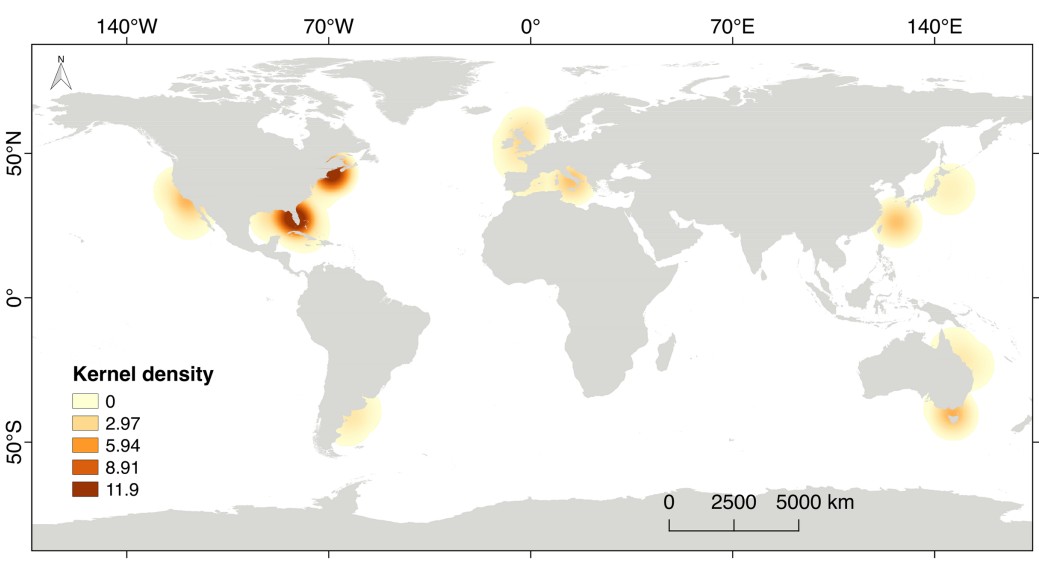

**Figure 7 Kernel's density heat map indicating the regions with scientific effort related to steroid hormones and reproduction in chondrichthyans during 1963–2020 (*n* = 59 papers).**

## CONCLUSIONS

The study of steroid hormones could become increasingly relevant for the science-based management and conservation of marine wildlife such as chondrichthyans. In recent decades, there has been an increase in the protection of sharks and rays worldwide, as well as of the habitats where they feed or reproduce (*Maestro et al., 2019*). The holistic science-based approach to the creation of marine protected areas should include the study of the reproductive biology of the species present in these ecosystems. This would benefit from steroid analyses, along with other methods such as ultrasounds and morphometrics for the biological monitoring of protected populations in critical habitats such as nursery areas (*Anderson et al., 2018*; *MacKeracher, Diedrich & Simpfendorfer, 2019*).

Steroid hormone analyses with traditional methodologies requiring dead specimens will allow the generation of baselines, particularly for the characterisation of steroid profiles throughout the reproductive cycle in the different strategies found in chondrichthyans (*Prohaska et al., 2013a*). Analyses of sex steroids besides E2, *T* and P4 are highly needed to bring to fruition all the benefits that hormonal investigation can provide to the field.

In the reviewed literature, the effects of steroid hormones on the reproductive biology of chondrichthyans were discussed according to sex and embryonic nutrition. However, most of these studies analysed three sex steroids and only a few publications detected and quantified more than such number. This can be partially explained by the limitation in tissue mass, financial support, available antibodies, and methods for the simultaneous analysis of a larger quantity of hormones using the same biopsy. Systematic research using captive model species under controlled environments can clarify the function of E2, *T* and P4 on specific biological phenomena, while the combination of methods could

provide relevant insights of the role of other steroid hormones during such reproductive processes.

Although the obtained information is highly valuable for present and future comparisons, the potential bias in terms of cross reactivity should be considered (*Nozu et al., 2015*, *2017*; *Mylniczenko et al., 2019*). Sharks, rays and chimaeras depose urea and other metabolic waste such as trimethylamine oxide in the blood (*Compagno, Dando & Fowler, 2005*; *Nelson, Grande & Wilson, 2016*). This constitutes an adaptation for energetics in terms of homeostasis, as blood from chondrichthyans has a higher salinity than water (*Compagno, Dando & Fowler, 2005*). In this regard, the presence of metabolites interacting with antibodies during immunoassays should be analysed in future studies, since most of the research used blood with no consideration of metabolic waste and pollutants that could be present in the bloodstream of these taxa.

The use of sensitive physical-based separation techniques that can simultaneously detect multiple steroid hormones and their metabolites in tissues other than blood, has proved to be useful in other groups of vulnerable species such as cetaceans (*Boggs et al., 2017*; *Hayden et al., 2017*) and in the round stingray *U. halleri* (*Lyons & Wynne-Edwards, 2019*). Although it is a new approach for the study of endocrinology in marine wildlife, LC-MS/MS has shown high-quality results (limit of detection, precision and recovery) using small tissue mass obtained from minimally invasive sample collection such as skin (*Boggs et al., 2017*; *Hayden et al., 2017*). The validation of this method in different matrices and species of chondrichthyans, along with complementary techniques, can create a new alternative for the study of commercial and protected species (*Hammerschlag & Sulikowski, 2011*; *Wudy et al., 2018*).

The study of hormone panels from small samples obtained by non-lethal collection using validated physical-based separation methodologies, could greatly benefit the scientific monitoring and protection of vulnerable species. One of the main concerns should be focused on the successful extraction, detection and quantification of steroid hormones from small biopsies of tissues (*Boggs et al., 2017*; *Hayden et al., 2017*). When it is achieved, these non-lethal techniques will be an important tool for research, and therefore the conservation of protected and cosmopolitan taxa such as the white shark, whale shark and mantas that are commercially used by ecotourism in several localities around the world (*Hammerschlag & Sulikowski, 2011*; *Cisneros-Montemayor et al., 2020*).

## ACKNOWLEDGEMENTS

The authors thank E. Georgina Molina and Leonardo Trejo for their support in the design of the figures.

### Funding

Edgar Eduardo Becerril-García and Daniela Bernot-Simon received a national scholarship from Consejo Nacional de Ciencia y Tecnología (MX). Marcial Arellano-Martínez and Felipe

Galván-Magaña received funding through grants from the Comisión de Operación y Fomento de Actividades Académicas, Estímulo al Desempeño de los Investigadores (IPN), and Sistema Nacional de Investigadores (SNI). Edgar Mauricio Hoyos-Padilla and Edgar Eduardo Becerril-García received funding from Ocean Blue Three and Fins Attached. Céline Godard-Codding received funding from Texas Tech University. The funders had no role in study design, data collection and analysis, decision to publish, or preparation of the manuscript.

### Grant Disclosures
The following grant information was disclosed by the authors:
Consejo Nacional de Ciencia y Tecnología (MX).
Comisión de Operación y Fomento de Actividades Académicas.
Estímulo al Desempeño de los Investigadores (IPN).
Sistema Nacional de Investigadores (SNI).
Ocean Blue Three.
Fins Attached.
Texas Tech University.

### Competing Interests
Edgar Mauricio Hoyos-Padilla is the General Director of Pelagios Kakunjá A.C. and a Scientific Advisor for Fins Attached.

### Author Contributions
- Edgar Eduardo Becerril-García conceived and designed the experiments, performed the experiments, analysed the data, prepared figures and/or tables, authored or reviewed drafts of the paper, and approved the final draft.
- Marcial Arellano-Martínez conceived and designed the experiments, authored or reviewed drafts of the paper, and approved the final draft.
- Daniela Bernot-Simon performed the experiments, analysed the data, prepared figures and/or tables, authored or reviewed drafts of the paper, and approved the final draft.
- Edgar Mauricio Hoyos-Padilla analysed the data, prepared figures and/or tables, authored or reviewed drafts of the paper, and approved the final draft.
- Felipe Galván-Magaña conceived and designed the experiments, authored or reviewed drafts of the paper, and approved the final draft.
- Céline Godard-Codding conceived and designed the experiments, authored or reviewed drafts of the paper, and approved the final draft.

### Data Availability
The raw data consists of the literature used in the article.

### Supplemental Information
Supplemental information for this article can be found online at http://dx.doi.org/10.7717/peerj.9686#supplemental-information.

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
