# Peer review of "Steroid hormones and chondrichthyan reproduction: physiological functions, scientific research, and implications for conservation"

_PeerJ, doi:10.7717/peerj.9686_

## Round 0.1 · original submission · Major Revisions

In this review, most mentions of methodological aspects, and biological aspects are insufficient, Furthermore, the data on possible effects of steroid hormones in literature were described for each hormone, but not for each biological phenomenon (for example, follicle development, ovulation, reproductive behavior, and so on). Therefore, readers cannot understand how each biological phenomenon is regulated by such hormones.

I would also suggest to add tables and/or figures to make the article more attractive to readers.

·

Basic reporting

no comment

Experimental design

no comment

Validity of the findings

no comment

Additional comments

Authors summarized steroid hormone levels in Chondrichthyans using 59 published literatures during 1963-2020. This review will be used for researchers focused on Chondrichthyans.
Comments are as follows:
1. The title looks too broad. This review is mostly focused on steroid hormone levels. The present reviewer cannot understand the “physiological effects” in the title.
2. Line 26: literatures?
3. Lines 44, 272 and throughout the text: need consistency to use oestrogen or estrogen
4. Lines 269-270: use small capital for “androstane”, hydroxypregnenolone
5. Line 290: Oestradiol or Estradiol
6. Lines 323, 394 and throughout the text: delete a space between 88.1 and %
7. Line 660: physiol should be Physiol
8. Line 671: scientific name of animals should be italic
9. Line 686: world should be World
10. Line 689: biochem should be Biochem
11. Line 754: delete this space (line)
12. Line 831: reproduction of vertebrates should be Reproduction of Vertebrates
13. Line 836: bull should be Bull
14. Line 838: textbook should be Textbook
15. Line 839: endocrinology should be Endocrinology
16. Line 873: brain research should be Brain Research
17. Lines 882, 894-895: title should be small capital
18. Lines 900-901: The Reproduction and Development of Shards, Rays and Ratfishes
19. Add a space between line 917 and 918
20. Lines 951-952, lines 970-971: use large capital for the book title
21. Line 974: Plos should be PLoS
22. Figure 3: better to add full name of each method in the legend.
23. Legends of Tables 2 and 4, line 6: 17beta-oestradiol? Dihydrotestosterone should be 5alpha-dihydrotestosterone, 11-ketotestosterone
24. Table 2, Line 8: androstane-3alpha, 17beta-diol?
25. Table 2, Line 9: hydroxypregnenolone, 11-dehydrocorticosterone

Reviewer 2 ·

Basic reporting

The authors addressed an important nitch, which was steroid hormones and the reproduction of chondrichthyans. The manuscript is fascinating and well-organized.

Experimental design

The authors focused on percentages of studies covered certain steroid hormones. However, it is more important to discuss data quality, such as replicates and assay specificities.

Validity of the findings

The authors should describe their thought based on previous data rather than summarizing thought by previous studies.

Additional comments

There are a couple of points to improve the manuscripts. Please find them in an attached PDF.

Annotated reviews are not available for download in order to protect the identity of reviewers who chose to remain anonymous.

Reviewer 3 ·

Basic reporting

In this review article, the authors studied literatures on cartilaginous fish reproduction with special reference to sex steroid hormones. This study covers a total 59 peer reviewed scientific papers from 1963-2020, and analyzed with regard to species investigated, methodologies and steroids investigated, which is unique and provides important information to this scientific field.

Experimental design

This review article contains four major analyses, namely, analyzed species, methods and biological matrices, steroid hormones and reproductive biology, and scientific research. The first three analyses are clear, but the section “scientific research” seems not to be clear in terms of the purpose of this study. In the “steroid hormones and reproductive biology” section, the data in literatures were well described for each steroid, but it was not easy to understand systematic contribution of multiple steroids to biological actions (phenomena). I strongly suggest that, in addition to the data for each steroid, the authors should analyze data in literature with regard to each biological phenomenon (such as follicle development, ovulation, reproductive behavior, and so on) and describe possible contribution of steroids to each phenomenon (if necessary, with additional table). I am afraid that the readers cannot understand how each biological phenomenon is regulated by steroid hormones. Discussion with other vertebrates, in particular with teleost fish, would also be required.

Validity of the findings

As described above, the analyses are unique and provide important information to the scientific field, but more analysis on biological function of steroid hormones are necessary. Currently, looks like just reviewed methodology, more biological values are necessary.

Additional comments

1. The authors mention that “there is no general pattern that could describe the specific actions of sex steroids in cartilaginous fish” (lines 115~, and others). For this reason, the authors mentioned such as diversity of species, variety of embryonic nutrition strategies, paucity of species investigated, and so on. I agree these in part, but more serious problem in the cartilaginous fish research would be the lack of comprehensive (detailed, systematic) research using captive model species.
2. line 135; “improved our understanding of the endocrinology”. Please mention more in detail.
3. lines 228-230; “In the case of blood, the original volume per sample ranged between 0.5-20 mL, while the volume for analysis was 50-1000uL”. In Table 2, the volume for analysis should be included in addition to the volume per sample. The authors discuss about the sample volume (for example, lines 235-239 and 532-533) that the methods have generally require a large sample mass for assays. Is this correct? In particular for RIA and EIA, the plasma sample volume required for E2, T, P4 would be less than 0.5 mL in total.
4. lines 246-249 and others; did steroid hormone levels in skeletal muscle show parallel changes with plasma levels?
5. The authors described about corticosterone as a corticosteroid. However, in cartilaginous fish, 1alpha-hydroxycorticosterone (1alpha-OH-B) is a major corticosteroid, and the levels of 1alpha-OH-B in plasma seem to be 100-times higher than the levels of corticosterone in a recent study (Iki et al., Gen. Comp. Endocrinol., 292, 113440, 2020). The authors should at least state in the text that the major corticosteroid of cartilaginous fish is 1alpha-OH-B.
6. The authors frequently mentioned in the text that “most studies measured only three steroids (E2, T and P4), and the limited number of studies reporting other steroids”, and this is one reason why biological actions of steroid hormones in reproduction have not been well clarified in cartilaginous fish. I agree in part, but function of E2, T and P4 have not been clarified yet. Again, the lack of comprehensive (detailed, systematic) research using captive model species would be the most serious problem.
7. line 606; “potential bias from immunoassays”, unclear.
8. The Conclusion section seems to be repetitive. For future perspectives, biological aspects such as “how to clarify biological actions of steroid hormones in cartilaginous fish reproduction” seem to be lacking.
9. This reviewer consider “sex steroid (hormone)” is appropriate than “sexual steroid (hormone)”.

---

## Round 0.2 · Minor Revisions

Your manuscript has been deeply improved. However, some more information about the extraction method in materials and methods is needed. Other minor comments are marked in the annotated manuscript by reviewer 2.

·

Basic reporting

no comment

Experimental design

no comment

Validity of the findings

no comment

Additional comments

Authors adequately revised the manuscript following comments from reviewers. No further comments on this manuscript.

Reviewer 2 ·

Basic reporting

The authors addressed all my previous comments. Here are all the minor comments. Please find them in an attached PDF.

Experimental design

The authors need to describe more about the extraction method. To compare the extraction efficiencies, it is critical whether extraction methods were the same or different. Also, describe what "-" indicates in Table 2.

Validity of the findings

no comment

Additional comments

Overall, very informative. This manuscript will help to find where the gaps are for future studies.

Annotated reviews are not available for download in order to protect the identity of reviewers who chose to remain anonymous.

Reviewer 3 ·

Basic reporting

No more comment

Experimental design

No more comment

Validity of the findings

No more comment

Additional comments

The authors have adequately addressed all the comments. The manuscript has been sufficiently improved; in particular, newly added Figures 5 and 6, and descriptions concerning current whole picture on regulation of reproductive phenomena by steroid hormones are highly meaningful.

---

## Round 0.3 · accepted · Accept

You have made all the modifications/suggestions indicated previously in order to improve your manuscript and I'm pleased to inform you that now it can be accepted for publication in PeerJ.

Thank you for submitting your work to this journal.